# Combining interventions to reduce the spread of viral misinformation

Joseph B. Bak-Coleman [1,2,3] ✉, Ian Kennedy [1,4], Morgan Wack [1,5], Andrew Beers [1,6], Joseph S. Schafer [1,6], Emma S. Spiro [1,3,4], Kate Starbird [1,7] and Jevin D. West [1,3]

**Misinformation online poses a range of threats, from subverting democratic processes to undermining public health measures. Proposed solutions range from encouraging more selective sharing by individuals to removing false content and accounts that create or promote it. Here we provide a framework to evaluate interventions aimed at reducing viral misinformation online both in isolation and when used in combination. We begin by deriving a generative model of viral misinformation spread, inspired by research on infectious disease. By applying this model to a large corpus (10.5 million tweets) of misinformation events that occurred during the 2020 US election, we reveal that commonly proposed interventions are unlikely to be effective in isolation. However, our framework demonstrates that a combined approach can achieve a substantial reduction in the prevalence of misinformation. Our results highlight a practical path forward as misinformation online continues to threaten vaccination efforts, equity and democratic processes around the globe.**

Misinformation has become a pervasive feature of online discourse, resulting in increased belief in conspiracy theories, rejection of recommended public health interventions and even genocide[1,2]. Academics and those working in industry have proposed a host of potential solutions, ranging from techniques for detecting and removing misinformation to empowering users to be more discerning in their sharing habits[3–5]. Despite an abundance of proposed interventions, online misinformation remains a global problem[6,7]. For instance, the 2020 US Presidential election and subsequent insurrection at the Capitol building highlighted how pervasive online misinformation can lead to real-world harm.

Real-world violence occurred as a result of a broader narrative that questioned the election's legitimacy, which arose from a series of more specific claims. Most claims were characterized by a brief period (that is, hours or days) of rapid growth in discussion and sharing[2]. Some of these incidents quickly died out, while others had multiple waves, spread to other platforms and often became consolidated into broader narratives. Early response to rapidly spreading misinformation provides a source of promise for successful intervention, as disrupting viral spread may have cascading effects on narrative consolidation and future engagement. Unfortunately, the rapid growth inherent to viral misinformation makes it challenging to assess and respond to in a timely manner. Effective intervention by platforms and policymakers requires a temporally aware framework for the quantitative comparison of proposed interventions.

Lacking such a framework, it is unclear whether existing strategies are sufficient to produce meaningful results. Crude approaches such as outright removal and banning of content or accounts will certainly work if applied in excess, yet they come with costs to freedom of expression and force private entities to be arbiters of truth. For judicious use, questions arise about how soon and how much removal is necessary for a meaningful effect. Similarly, interventions that rely on empowering individuals to consume and share more discerningly have shown promise in experimental contexts. Still, it remains unclear what impact they will have at scale[4].

Beyond comparison, we lack an understanding of when—and indeed whether—multiple interventions can act synergistically to reduce the spread of misinformation. Unfortunately, experiments do not adequately address these questions, as their efficacy at scale cannot be directly inferred. For example, the consequences of variation in follower counts, which span eight orders of magnitude, would be difficult to capture in the lab. Moreover, the unique behaviour of highly influential repeat spreaders (for example, coordination and early amplification) will certainly impact dynamics and thus efficacy[2]. Furthermore, the timescales at which viral misinformation events occur online (that is, hours, days or weeks) pose challenges to extrapolation from comparatively brief experiments.

Platforms such as Twitter can and do run experiments at scale, yet the data and methods are not generally made available for open research[8]. Moreover, private companies conduct these experiments in the absence of scientific discourse or ethical oversight. While we can observe some changes in response to platform policies, it is nearly impossible to disentangle the effect of the policy from unseen algorithmic alterations, interface modifications or behavioural changes[9].

Despite these challenges, insight is urgently required, as poorly implemented policies or inaction could exacerbate misinformation and cost lives[10]. Towards this goal, we derived and parameterized a generative model of misinformation engagement (that is, the total discussion and sharing of posts related to false information) using a large corpus of Twitter posts collected during the 2020 presidential and congressional elections in the United States[2]. This approach captures the dynamics of viral misinformation at scale and across time in a real-world context. We relied on this model to examine the efficacy of misinformation interventions both in isolation and when deployed in combination. Finally, we examined how the spread of misinformation during viral periods impacts subsequent engagement.

[1]Center for an Informed Public, University of Washington, Seattle, WA, USA. [2]eScience Institute, University of Washington, Seattle, WA, USA. [3]The Information School, University of Washington, Seattle, WA, USA. [4]Department of Sociology, University of Washington, Seattle, WA, USA. [5]Department of Political Science, University of Washington, Seattle, WA, USA. [6]Paul G. Allen School of Computer Science and Engineering, University of Washington, Seattle, WA, USA. [7]Human Centered Design and Engineering, University of Washington, Seattle, WA, USA. ✉e-mail: joebak@uw.edu

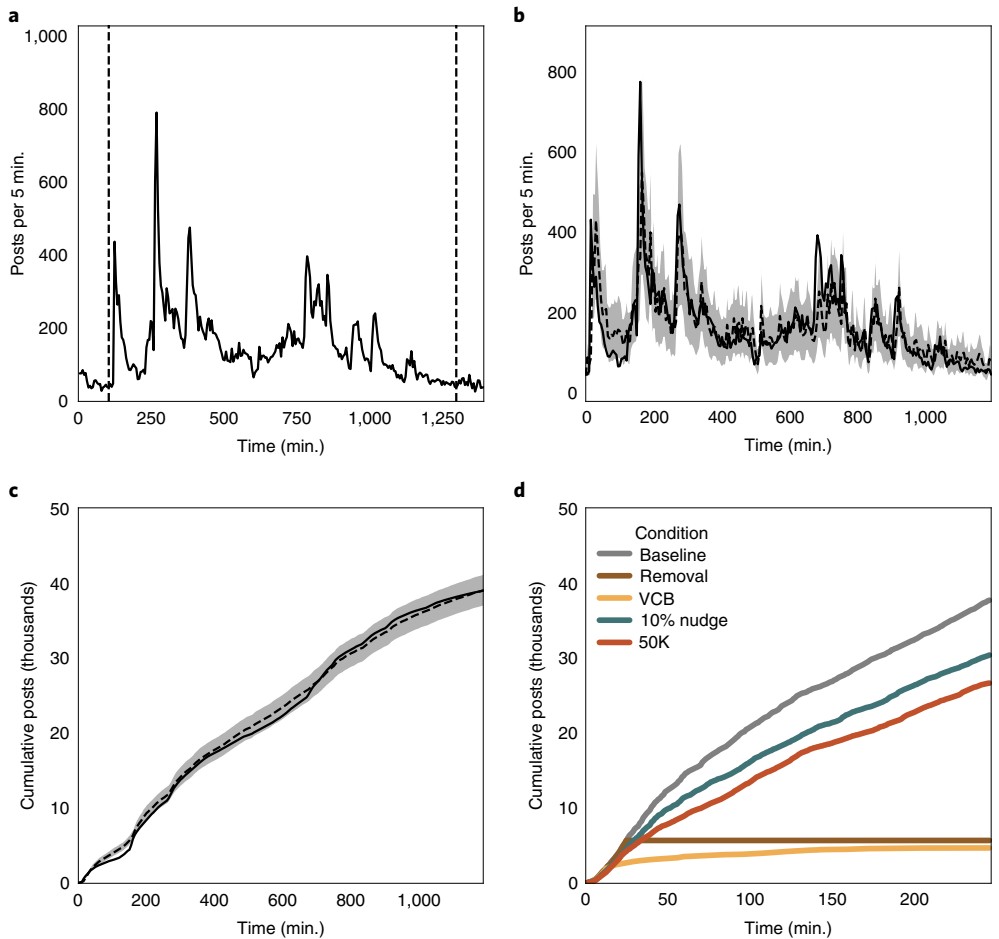

**Fig. 1 | Overview of our data-processing and generative model of viral misinformation spread. a**, Example of an event segmented from a larger incident (dashed lines). **b**, Our model fit to the time series for a single event. The dashed line indicates the expected value; the shaded region denotes the 89% CI. **c**, Cumulative engagement as a measure of total misinformation. The lines and shading are as in **b**. **d**, Model-simulated platform interventions for a single event. The lines indicate median cumulative engagement over 100 simulations. Grey indicates the baseline, blue indicates a 10% 'nudge', orange indicates banning, yellow indicates a virality circuit breaker and brown indicates the outright removal of content.

## Results

**Data and model overview.** Our analysis relies on a dataset of Twitter posts collected during the 2020 US election. This dataset was extracted from a broader set of 1.04 billion election-related posts collected between 1 September 2020 and 15 December 2020. To construct our dataset, we first identified 430 incidents—distinct stories that included false, exaggerated or otherwise misleading claims or narratives. Search terms were devised for each incident, extracting 23 million posts generated by 10.8 million accounts from the broader collection. Search terms and incidents were identified through real-time monitoring and updating by dozens of analysts and several community partners as part of the Election Integrity Partnership[2]. As such, we believe that our dataset provides a thorough—if not comprehensive—overview of misinformation during the 2020 US presidential election.

From each incident's time series, we extracted events, defined as periods in which a story exhibited rapid growth and decay (Methods and Fig. 1a). This process identified 544 potentially viral events, including 14.6 million Twitter posts (tweets, retweets, replies and quote tweets). The number of viral events (544) is higher than the number of incidents (430) because an incident could have more than one viral event. We then derived a generative model of viral information spread. Our model was adapted from models of super-spreading in infectious disease, allowing us to characterize the spread of misinformation and the efficacy of interventions.

Our model treats virality as temporally varying, increasing proportionally to the out-degree (that is, the number of followers) of the accounts that post about a topic. We also assume that virality decays over time as the network saturates and new topics arise. The details can be found in the Methods.

Using Bayesian methods, we estimated model parameters for each event (Fig. 1b,c). As our model is not expected to fit all topics discussed during the election, we developed inclusion criteria to ensure that our model was appropriate for a given event and that the derived parameters could reproduce the observed engagement (Methods). This led to our final dataset of 10.5 million posts from 454 events. We then simulated total engagement by seeding the model with the estimated parameters, posts in the initial five-minute time step and the empirical distribution of follower counts for each five-minute interval. Our simulated engagement strongly corresponded to the observed engagement for events spanning several orders of magnitude in post volume (Supplementary Fig. 2). For the results presented throughout, we modified our model in various ways and evaluated the total simulated engagement across all included events (Fig. 1d and Methods).

**Fact-checking and time-lagged approaches.** We begin by considering the impacts of interventions targeting specific content on user engagement (that is, total posts: retweets, tweets, quote tweets and replies). Among the more commonly employed strategies during

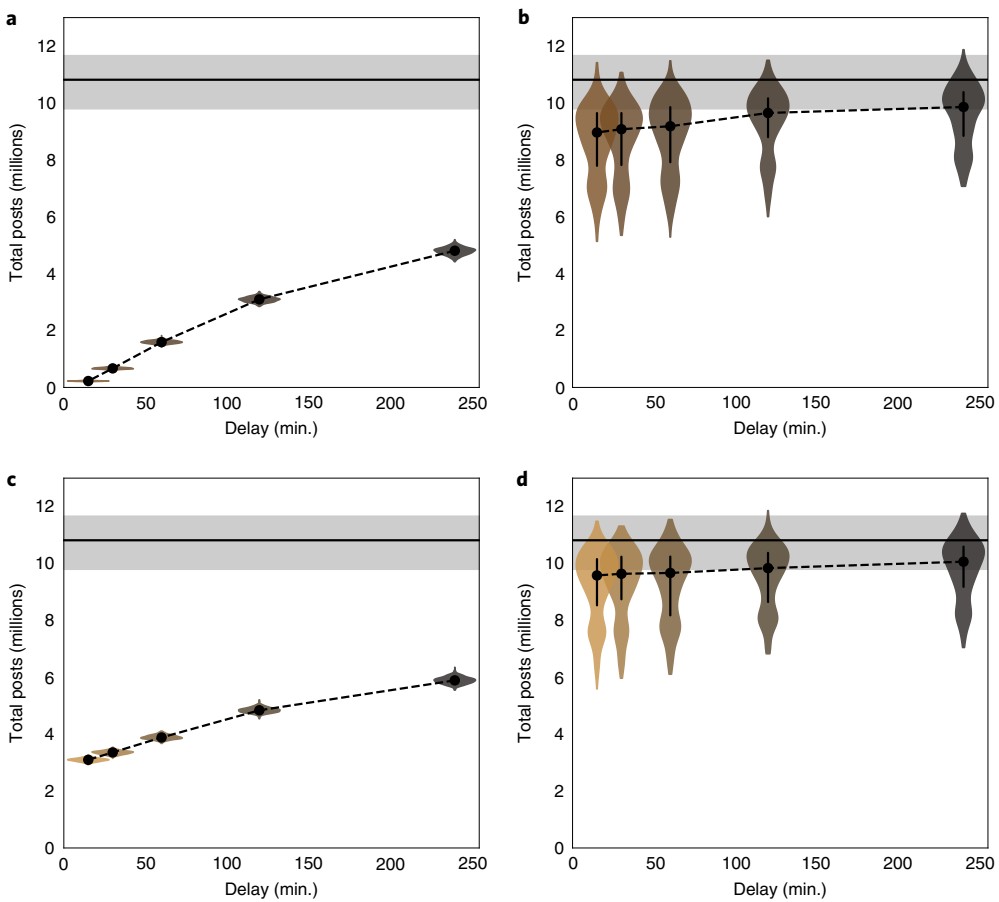

**Fig. 2 | Simulated impacts of content removal and virality circuit breakers. a**, The impact of the outright removal of all misinformation-related posts following a delay specified in minutes. **b**, As in **a** if only 20% of events are removed. **c**, The impact of applying a virality circuit breaker that reduces virality by 10% to all misinformation events after a specified period. **d**, As in **c** if the virality circuit breaker is applied to only 20% of events. The horizontal grey bars in each plot represent the baseline conditions, with the line indicating the mean and the shaded area highlighting the 89% CI. The violins indicate the simulated distribution of total posts across all events. The dots and lines within the violins indicate the median and interquartile range.

the 2020 US election was identifying specific misinformation and taking action, ranging from applying a label to outright removal[2]. These approaches share a common feature of requiring time before action is taken. Time is necessary not only to identify misinformation but also to decide on an appropriate response.

In an extreme case, a platform could remove or hide all content matching search terms related to an emerging misinformation incident. To simulate this, we ran our model until a given time point at which growth was stopped entirely (Fig. 2a). Our results indicate that outright removal can indeed be effective, producing a 93.8% median reduction in total posts (that is, tweets, replies, quote tweets and retweets) on the topic, if implemented within 30 minutes (89% credible interval (CI), (92.9, 94.4)). Even with a four-hour delay, our model indicates reductions of 55.6% (89% CI, (50.7, 59.2), Supplementary Table 1). These effects generously assume that platforms can monitor, detect, sufficiently fact-check and implement a full removal response within the specified time frame. As such, the efficacy is dramatically reduced if only a fraction of events lead to action (Fig. 2b and Supplementary Table 2).

A more plausible approach could involve 'virality circuit breakers', which seek to reduce the spread of a trending misinformation topic without explicitly removing content—for example, by suspending algorithmic amplification[11]. This approach allows platforms to consider ethical ramifications while minimizing the public relations challenges accompanying direct forms of action. This could aid in lowering the threshold for fact-checking and therefore

enable quicker response times. We simulate the impact of virality circuit breakers by proportionately reducing the latent virality parameter in our model after a fixed time interval.

Through simulations, we reveal how virality circuit breakers can have similar efficacy to outright removal even if the amount by which virality is reduced is small (Fig. 2c and Supplementary Table 3). For instance, a 10% reduction in virality implemented four hours after the start of an event can reduce the spread of misinformation by nearly 45.3% (89% CI, (39.4, 49.8)). As with outright removal, however, the efficacy is primarily limited by the proportion of events for which the platforms take action (Fig. 2d and Supplementary Table 4).

**Nudges and reduced reach.** A drawback of fact-checking-based approaches is that they are most applicable to transparently false or readily falsifiable claims[2]. Many instances of misinformation involve claims that are partly true or require non-trivial time to debunk. Claims that there are statistical irregularities in reported vote tallies, for example, require a statistician gathering and analysing the data and determining merit. Depending on the implementation, time-lagged responses may require users do not devise workarounds (for example, posting screenshots or off-platform links).

These challenges motivate approaches that leverage individual discretion to reduce the spread of misinformation[5]. For instance, encouraging users to consider accuracy has been shown to improve discernment of false information by 10–20%[4]. This can be

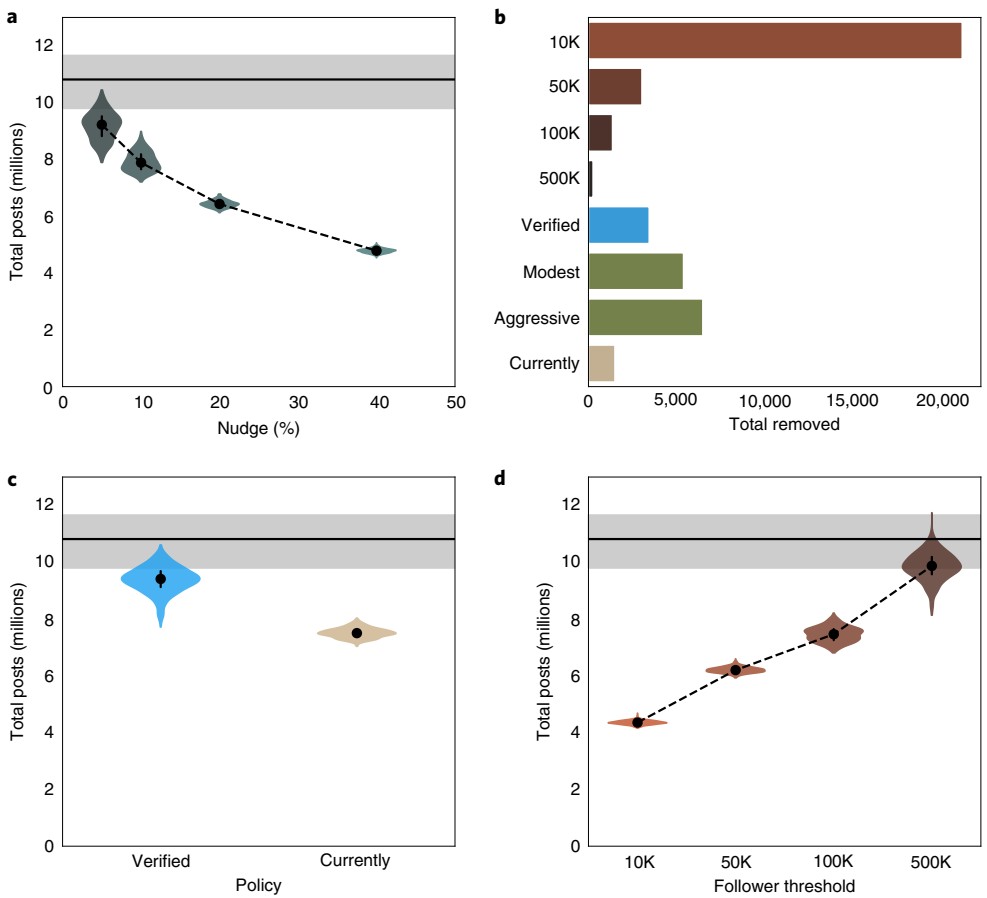

**Fig. 3 | Simulated efficacy of nudging and account removal interventions. a,** The effect of nudges that inoculate a percentage of the population against spreading misinformation. Shown is the cumulative total engagement across all events. **b,** Number of accounts that either are currently removed or would have been removed under a three-strikes policy. Brown bars indicate the threshold being applied to all users above a given threshold of followers in thousands, K. **c,** The effect of account removal for those that are currently banned (orange) or those banned following a three-strikes rule applied solely to verified accounts (blue). **d,** As in **a** and **c**, but showing the impact of enacting three-strikes policies with varying thresholds. The horizontal grey bars in **a**, **c** and **d** represent the baseline conditions, with the line indicating the mean and the shaded area highlighting the 89% CI. The violins indicate the simulated distribution of total posts across all events. The dots and lines within the violins indicate the median and interquartile range.

implemented by warning users when they encounter potentially false or misleading information, but this still requires the labelling of that content as false, as does fact-checking.

A central question is whether a modest reduction in individual sharing behaviour can lead to a more dramatic change in overall rates of misinformation. Agent-based models support this notion across a range of network topologies[4]. From the perspective of our model, nudge-based approaches can be simulated by maintaining the parameters from the initial model fit while proportionally reducing the following of every user that discusses an incident.

Using our model to simulate nudges, we find that they can indeed reduce the prevalence of misinformation (Fig. 3a and Supplementary Table 5). Nudges that reduce sharing by 5%, 10%, 20% and 40% result in a 15.2% (3, 24.3), 26.4% (18.1, 33.5), 40.3% (34, 45.1) and 55.6% (50.9, 59.0) reduction in cumulative engagement, respectively (Supplementary Table 5). The median effect tends to be larger than the nudge, suggesting a degree of feedback whereby the individual effect of a nudge is compounded in the misinformation dynamics.

**Account banning.** In our dataset, several accounts shared or amplified misinformation across multiple incidents[2]. Moreover, some of these repeat offenders had outsized audiences compared with the average Twitter user—ranging from hundreds of thousands to

millions of followers. While the removal of repeat offenders during the election was rare, several were removed after the violent insurrection at the US Capitol on 6 January 2021. The question remains whether removing these accounts, or account-focused policies in general, would have a meaningful impact on misinformation. While large followings often confer engagement, it remains possible that there is sufficient sharing from smaller accounts to ensure the spread of misinformation even in the absence of the larger removed accounts[12].

One challenge in modelling account removal is that there are probably non-trivial relationships between account size, the propensity to share misinformation and the timing at which certain accounts amplify narratives. A large account that regularly shares misinformation in the first five minutes will have an outsized effect compared with a smaller account that occasionally shares misinformation hours later. To account for this, our model samples from the empirical follower-count distribution in a given time step. Furthermore, as the identities of individuals are known, we can remove specific accounts and simulate total engagement (Methods). In other words, our simulations are conditioned on unseen patterns of and variation in individual behaviour without explicitly quantifying the differences in individual behaviour. Our model and simulations therefore exhibit robustness to considerable unmeasured real-world complexity.

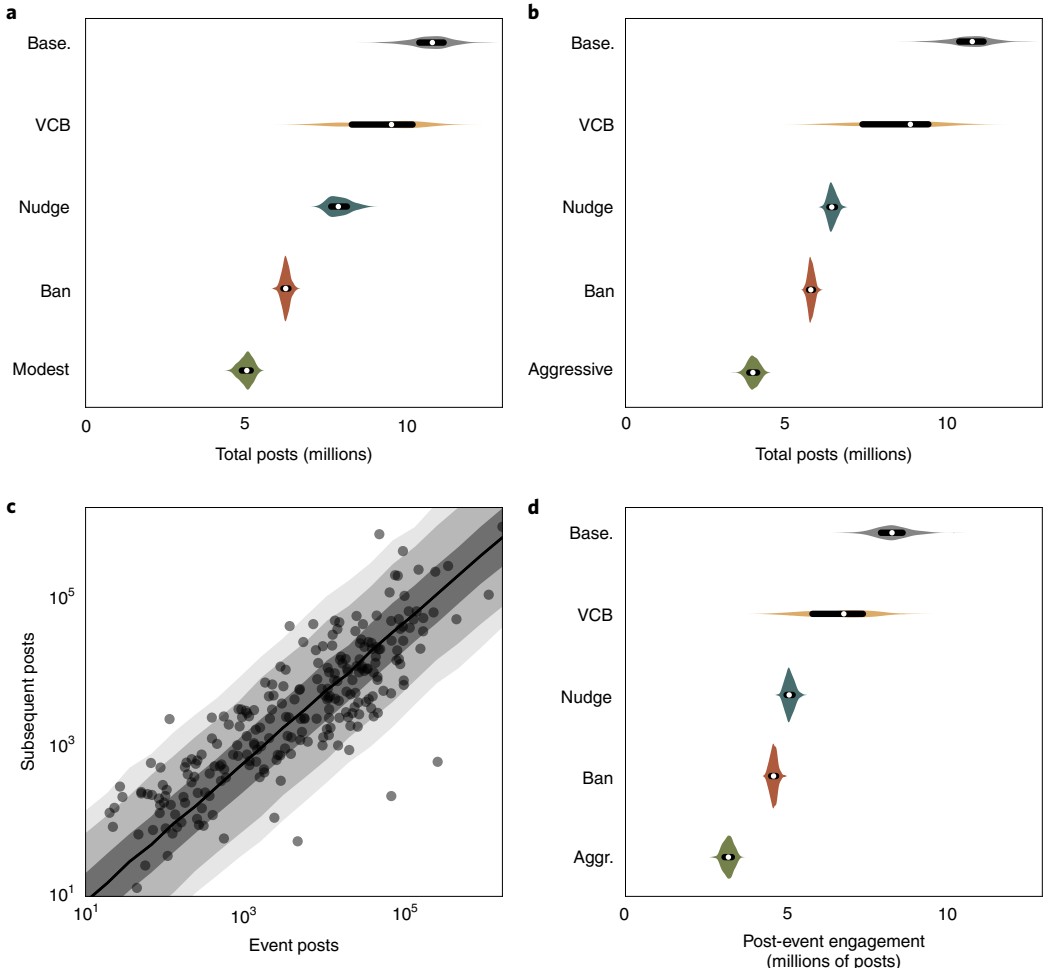

**Fig. 4 | Simulated impact of combining interventions. a**, The impact of a modest combined approach to intervention (described in the text, green) and each intervention applied individually. **b**, The impact of a more aggressive combined approach (described in the text, green) and each intervention applied individually. **c**, Relationship between engagement within the largest viral event for a given incident and subsequent engagement. **d**, Expected post-event engagement given the action taken during an event. The dots and lines within the violins indicate the median and interquartile range.

We first consider the consequences of account removals ($N = 1,504$) in early 2021. These accounts were identified by examining accounts with posts in our dataset that could not be retrieved with an API call in late January. We seek to answer whether previously implemented account removal is sufficient to curb misinformation going forward. Our simulations reveal that the removal of these accounts from our dataset reduces total engagement with misinformation by 30% (89% CI, (10.2, 22.7), Fig. 3c). This is comparable in efficacy to a 10% reduction in the sharing of misinformation (that is, a nudge) impacting all accounts in the absence of removal.

We next consider a 'three-strikes' rule in which accounts are removed from the platform after they are detected in three separate incidents of misinformation (that is, topics, regardless of the number of posts on a given topic). For these simulations, any interaction with or amplification of misinformation (that is, tweets, retweets or quote tweets) would be counted as a strike. A policy focused solely on original content could be gamed by using large accounts to amplify smaller disposable accounts. This type of policy would avoid banning accounts that were swept up by a given piece of misinformation and repeatedly tweeted while focusing on those that spread misinformation more broadly. When the policy is applied solely to verified accounts, we observe a 12.7% drop in cumulative engagement (89% CI, (2.3, 23)), which likewise is similar in efficacy to a small nudge rolled out across the board (Supplementary Table 6

and Fig. 3c). If, instead of verification, the policy is applied on the basis of the number of followers an account has, pronounced effects are observed only when the threshold is quite low (~10,000 followers), requiring large numbers of accounts to be removed (Fig. 3b,d and Supplementary Table 7).

**Combined approaches.** All of the approaches above exhibit some efficacy in reducing engagement with viral misinformation. Unfortunately, each strategy tends to become maximally effective in impractical regions of parameter space. The outright removal of misinformation is particularly effective, yet it is difficult to imagine that more than a small fraction of misinformation can be easily removed. Virality circuit breakers face similar challenges, albeit to a lesser extent. For nudges that minimally impact user experience yet improve individual discretion, effects far beyond ~20% are unlikely without a major breakthrough in information literacy or social psychology[4]. In the case of banning specific accounts, low follower thresholds increase the number of accounts removed, and thus the costs and challenges, super-linearly.

We therefore consider a combined approach relying on only modest implementations of each of the strategies studied above. Specifically, viral circuit breakers are employed for 5% of content, reducing virality by 10%, and enacted after 120 minutes. Of the content subjected to a viral circuit breaker, 20% is subsequently

removed outright after four hours. We further assume a 10% reduction in individual sharing of misinformation resulting from a nudge. Finally, accounts that have been removed remain banned, and a three-strikes policy is applied to verified accounts and those with more than 100,000 followers. Our model reveals that even a modest combined approach can result in a 53.3% (89% CI, (48.2, 58.2)) reduction in the total volume of misinformation (Fig. 4a and Supplementary Table 8).

We additionally consider a more aggressive version of a combined policy, applying viral circuit breakers to 10% of content and reducing virality by 20% while cutting response times in half. We further assume a 20% nudge and reduce the threshold for the three-strikes policy to 50,000 followers. This more aggressive approach dramatically reduces misinformation by 63.0% (89% CI, (58.4, 66.9); Fig. 4b and Supplementary Table 9). Similar efficacy from standalone approaches would either be impossible or require draconian removal of content and accounts.

One limitation of our model is that it relies on assumptions specific to periods of viral misinformation spread. In our dataset, only 40% of posts occur during the largest event for a given incident. Yet 25% of engagement occurs after the largest event. The remaining posts are either smaller events prior to the largest event or low-volume posts picked up by our search terms prior to the largest event. While our model cannot provide direct insight into how interventions will impact engagement during these periods, we can gain indirect insight by considering the relationship between the size of an event and subsequent discussion.

Our data demonstrate that the size of an event is strongly predictive of subsequent engagement (Fig. 4c; Bayesian log-normal regression $\beta = 0.94$; 89% CI, (0.92, 0.95); Supplementary Table 10). Using this relationship, we can estimate subsequent discussion on the basis of simulated, intervention-adjusted engagement during the largest event (Methods). Through this method, we reveal that the impact of interventions on post-event engagement is likely to be similar in magnitude to the efficacy during an event (Fig. 4d and Supplementary Fig. 1).

## Discussion

Our derived model, grounded in data, provides quantitative insight into the relative efficacy of proposed interventions. We reveal through simulation that proposed interventions are unlikely to be effective if implemented individually at plausible levels. Effective removal of content or virality circuit breakers would require large teams and rapid turn-around times and would place content decisions squarely in the hands of private organizations. Nudges are promising but unlikely to be a panacea at known levels of efficacy[4]. Banning accounts seems to be the most workable solution but would require the removal of tens of thousands of users to be effective.

However, our results show that combining interventions at plausible levels of enforcement can effectively reduce the spread of viral misinformation. While it is unsurprising that multiple interventions outperform individual approaches, our findings provide insight into the magnitude of that difference. The efficacy of a combined approach depends not only on the nature of individual interventions but also on how they interact with one another, the dynamics of misinformation spread, the event duration, user sharing behaviour, user follower counts and how these factors change throughout a disinformation campaign. In fitting our model to a large corpus of events during an active period of mis- and disinformation, our results are conditioned on much of this complexity. Furthermore, by drawing from the empirical distribution of users' follower counts, our model indirectly and implicitly accounts for unseen behavioural patterns of users and changes to their follower counts over time.

Our theoretical approach is limited in several key ways. Most notably, practical and ethical challenges preclude experimental validation of our theoretical findings. As our model evaluates the spread of viral misinformation at scale, experimental validation would require Twitter to adopt these interventions and apply them to millions of users. Moreover, limited transparency regarding the interventions used by Twitter makes it possible that some of the simulated interventions were in place, and our simulations reveal the benefit of increasing those interventions beyond their implemented amount[9]. Increased transparency on the part of platforms will be critical for future research on estimating the efficacy of interventions.

What further remains unclear is how changes in the magnitude of events will impact the longer-term dynamics of misinformation and translate to a reduction in harm. If implemented in tandem, multiple interventions may prove a sufficient shock to collapse the misinformation ecosystem altogether, as shock-induced collapse is a central feature of complex systems[13]. For instance, subsequent events probably depend on the size of previous events, and breaking that feedback could lead to greater gains than expected. However, this same body of literature suggests that an insufficient shock may yield only short-term changes as the system re-organizes and adapts.

We note that the results presented here rely on a simplified model of events on a single platform in a highly complex, multi-platform system. These types of simplifications are an inherent limitation of any approach—short of risky, large-scale experimentation. However, abstract models of complex processes have proved essential to predicting the benefits of interventions on complex systems, from mitigating the spread of disease to stabilizing ecosystems[14,15]. Models provide particular utility when experiments are unethical and impractical and the costs of inaction are high. Given the substantial risks posed by misinformation in the near term, we urgently need a path forward that goes beyond trial and error or inaction. Our framework highlights one such approach that can be adopted in the near term without requiring large-scale censorship or major advances in cognitive psychology and machine learning.

## Methods

**Data collection and processing.** All data were collected in accordance with the University of Washington Institutional Review Board. Our dataset was collected in real time during the 2020 US election. We relied on a set of 160 keywords to collect posts from Twitter's API (1.04 billion). The keywords were updated in response to new narratives—for instance, adding 'sharpiegate' and related terms after false narratives emerged about the use of Sharpie markers invalidating ballots. Working with the Electoral Integrity Partnership, we catalogued instances of false or misleading narratives that were either detected by the team or reported by external partners[2]. This led to a large corpus of tickets associated with validated reports of misleading, viral information about election integrity.

Tickets that shared a common theme were consolidated into incidents. We developed search terms and a relevant date range for each incident to query posts from our tweet database. Incidents ($N = 430$) were generally characterized by one or more periods of intense activity followed by returning to a baseline state (Fig. 1a). The search terms and descriptions of the incidents are provided along with the data.

We then wished to extract segments of the time series that exhibit macroscopic features consistent with viral dynamics. More specifically, candidate events should exhibit quiescent periods before and after the event where our search terms return to baseline levels. However, multiple peaks may occur between these boundaries. To extract candidate events, we computed the raw time series of post volume per five minutes for each of our distinct incidents. We then identified events by finding the five-minute interval within the aggregated time series with the largest number of collected posts. Other peaks in activity were considered part of separate events if they were at least 30% of the magnitude of the largest peak (to filter out noise). Starting with the largest peak, we identified its boundaries as the points before and after the peak where the number of posts in five minutes was less than 5% of the maximum volume. This may include multiple peaks within the same event, if no quiescent period occurred between them. We then repeated this process for all remaining peaks. If periods of activity less than 5% of the maximum peak height did not occur within the range of data collection, the first (or last) time point collected was used to denote the beginning (or end) of an event. Finally, events were required to last at least an hour (that is, 12 data points). This process extracted 544 candidate events from 269 incidents.

**Statistical and computational model.** *Model derivation.* We then derived a model of spreading dynamics during viral misinformation cascades. We restricted our model to the dynamics of misinformation flow within a single event rather than

longer-timescale processes such as the adoption of beliefs and behaviours. The spread of beliefs and behaviours often requires that multiple neighbours have adopted the state (that is, a complex contagion)[16]. The acceptance of a given misinformation narrative, for instance, can involve complicated cognitive processes involving partisan leanings, prior knowledge, attention, the message content and a host of other factors[4,17].

Re-sharing of information on Twitter, however, requires solely that a single neighbour has shared a piece of content for it to potentially be seen and retweeted[18,19]. Moreover, empirical work has demonstrated that out-degree (follower count) nearly linearly predicts engagement[20,21]. These features are hallmarks of simple contagions at the timescales of interest in our events. Following previous work, we therefore model the spread of viral misinformation as a simple contagion[22,23]. At the core of our model is a latent virality parameter, $v$, which tracks the amount of attention a topic is garnering over time. Unlike typical compartmental models, accounts vary widely in their out-degree from 0 followers to more than 100 million. In disease research, branching process models have incorporated various degree distributions to examine the role of super-spreaders[14].

We built on models of super-spreading and leverage the fact that the out-degree of each account can be estimated by their total followers[24,25]. When a user posts during an event, our model assumes that virality is increased proportionately to their number of followers (that is, the total exposed). However, network saturation and competition for attention with other topics can reduce virality over time. We incorporate this by adding a decay function, such that virality naturally decays over time. Together, growth from sharing and decay from saturation and competition define virality. Posts in a given time step are predicted by virality in the previous time step. These phenomena can be captured by a minimally parameterized branching process model, such that:

$$\mathbb{E}[y_t] = \exp(\alpha + \beta v_{t-1})$$
$$v_t = v_{t-1}\delta e^{-\lambda t} + x_{t-1}$$
$$x_{t-1} = \log\left(\sum_{j=1}^{y_{t-1}} F_{j,t-1} + 1\right)$$

(1)

where $y_t$ is the number of posts (that is, retweets, tweets, replies and quote tweets) at five-minute interval time $t$, $\alpha$ is the baseline rate of discussion and $\beta$ is the effect of virality, $v$. Virality is a latent parameter proportional to the total number of users at a given point in time that are exposed to misinformation. It represents the extent to which an event, at a given point in time, is visible in timelines across Twitter. Virality decays as an exponential function via $\delta$ and $\lambda$. Here, $\delta$ captures the baseline rate of decay per time step, and $\lambda$ controls how that decay changes over the lifetime of an event. This could be due to algorithmic processes favouring new content or user saturation for very large events. Every time step, for each of $y_t$ accounts that posts, the log sum ($x_t$) of their followers, $F_j$, is added to virality.

We note that our model does not explicitly incorporate a network, as is common in many simulations of information and behaviour spread online[16]. Our primary reason for doing this is that algorithmic filtering of content renders the true network topology unknown. Reconstructing a network would require additional epistemic assumptions, which could bias the results in opaque ways[26]. Moreover, research on disease has highlighted the utility of modelling interventions in the absence of network structure, notably when the degree distribution is known or approximated[14]. We note that the success of simple models in understanding the spread of infectious disease is not due to simplistic contagion dynamics. For disease, daily interactions, immune-system dynamics, population structure, behaviour and air-flow patterns create remarkably complex and dynamic network topologies of disease spread.

Our model was fit to each event using PyStan v.2.9.1.1 (refs. [27,28]). We fit events separately (rather than hierarchically) as they varied widely in their timescales, magnitudes and contexts within the broader 2020 election cycle. Of the 544 candidate events, our model performed well on 454 events (~10.4 million posts) of rapid misinformation spread. Our model was unlikely to be suitable for all events because it assumes that post volume is well predicted by the number of previously exposed accounts on Twitter. If, for instance, an incident received substantial news coverage (for example, Dominion software narratives), our model would probably fail.

To safeguard against this, we relied on a number of criteria to ensure model fit to a given event. Events were included in the final analysis if (1) the posterior 89% CI of total posts contained the observed value, (2) the chains successfully converged for all parameters ($\hat{R} < 1.1$), (3) the fit did not contain divergent transitions and (4) the event lasted longer than an hour (that is, >12 data points to fit). Other than these criteria, events surrounding the Dominion narrative were removed as they involved long periods of high-volume online discussion. This filtering process resulted in the inclusion of 454 events (83% of total events) and ~10.4 million posts.

*Statistical model.* We derived parameters for our model statistically using a custom-written model in Stan[27]. Posts $y_t$ at time $t$ are assumed to be distributed as a gamma–Poisson mixture (that is, negative binomial) with expected value $\mu_t$.

A gamma–Poisson distribution was chosen because it allows for overdispersion of discrete events occurring in a fixed interval (here, posts). Specifically:

$$y_t \sim \text{NegativeBinomial2}(\mu_t, \phi) \text{ for } t = 2...T$$
$$\mu_i = \exp(\alpha + \beta v_{t-1}) \text{ for } t = 2...T$$
$$v_t = v_{t-1}\delta e^{-\lambda t} + x_{t-1}$$
$$\alpha \sim \text{Normal}(-3, 3)$$
$$\beta \sim \text{Normal}(0, 3)$$
$$\delta \sim \text{Beta}(1, 1)$$
$$\lambda \sim \text{HalfExponential}(1)$$
$$\phi \sim \text{HalfExponential}(1)$$
$$v_1 = x_1$$
$$x_{t-1} = \log\left(\sum_{j=1}^{y_{t-1}} F_j + 1\right)$$

Here $\alpha$ is the baseline rate of detection for related keywords, and $\beta$ is the effect of virality, $v$, on posts in a subsequent time step. Virality is calculated as a decaying function of $v_{t-1}$ and the log of the sum of account follower counts $F_j$ for posts in the previous time step. One follower is added to each user to avoid an undefined value in time steps with no followers. The log transform accounts for the link function (exp), transforming the linear model into an expected value for the negative binomial distribution. Given the wide range of possible event shapes, generic, weakly informative priors were chosen for all parameters. The models were fit using NUTS in PyStan with the default sampling parameters[27,28].

*Computational model.* Our computational model relied on the posterior distributions of parameters obtained from fitting our statistical model separately to each event. For each simulation, one sample was drawn at random from the posterior for a given event. At $t=1$, the model was initialized with the volume of posts and total exposed users from the first time step in which any posts were observed. At each subsequent time step, our computational model predicted the number of new posts, $y_t$, by sampling from a negative binomial distribution as per our statistical model. For each of $y_t$ new posts, we drew a follower count from the actual distribution of accounts that retweeted for that event at that time step. Doing so allowed us to control for the possibility that some accounts tend to appear earlier in a viral event. This process was repeated for the duration of the actual event.

We simulated the removal of misinformation by simply setting $y_{t+1} = 0$ after at a specified intervention time, $t$. Virality circuit breakers were enacted by multiplying virality at each time step by a constant. For example, a 10% reduction in virality was implemented as $\hat{v}_t = v_t(1 - 0.1)$. As with content removal, this occurred only after a specified time step. In the case of the combined approach, virality circuit breakers (and subsequent removal) were employed at a given probability for each simulation run. We implemented nudges via multiplying follower counts by a constant, reducing the pool of susceptible accounts (that is, for account $j$, $\tilde{F}_j = F_j(1 - \eta)$). Finally, we implemented a three-strikes rule by identifying the third incident in which a given account appeared in our full dataset. They were removed from simulations for all events that occurred after their third strike.

Additionally, our model included a maximum value of twice the observed posts per time interval to account for a rare condition in which long-tail parameters would lead to runaway. This was observed to occur rarely enough to be challenging to quantify (<1% of model runs), but it was implemented to reduce upward bias in control conditions. We did this to ensure conservative estimates of efficacy, as interventions could reduce the possibility of runaway without meaningfully impacting engagement. Such a feature would be expected in any model of a growth process with exceptionally long-tailed distributions of follower counts and spread at a given time step (that is, a negative binomial).

For the figures shown in the main text and the tables presented in the Supplementary Information, we ran 500 simulations of all 454 events. For each run, we computed the cumulative engagement. The 500 simulations were summed across runs, from which we calculated the medians and CIs. All simulations were implemented in Python[29] v.3.9.10.

*Model validation.* Some form of model validation strengthens any theoretical approach. As data-derived models of large-scale processes are uncommon in the social sciences, we offer some notes on validation and its limitations in this context. Ideally, our findings could be externally validated in an empirical setting. In our case, the gold standard would be to have Twitter implement our recommended policies in some locations but not others and examine subsequent engagement with viral misinformation.

Validation of this sort is both practically and ethically prohibitive. Ethically, the application of our theory to real-world social networks should occur after broader scientific scrutiny and not before publication. As these experiments impose actual

costs on the individuals impacted by platform policies, a complete evaluation by the scientific community is necessary to evaluate potential benefits and mitigate risks. Ethical challenges aside, such an experiment is impractical, as it would require Twitter to rewrite its platform guidelines and hire fact-checkers at our suggestion. To the extent that Twitter conducts internal experiments, observational validation by the scientific community (that is, natural experiments) is confounded by unseen changes in the user interface, algorithmic sorting, concurrent A/B testing or other aspects of the experiment that are not disclosed to researchers.

This is a problem inherent to any data-derived model of a complex system at scale. Climate models suggest that reducing greenhouse gases will slow climate change and highlight the relative efficacy of various approaches[30]. Yet empirical validation at scale would require convincing nations to experimentally reduce greenhouse gases alongside a control world where these policies are not applied. Similarly, an experiment involving altering conditions in an enclosed space may be consistent with data-derived models yet provide little additional insight[31]. Furthermore, there is no known orthogonal world in which models of anthropogenic disturbance can be externally validated. Nevertheless, models of greenhouse gas reduction remain our best hope at reversing climate change. A recent perspective has argued that similar approaches are probably necessary for the stewardship of our social systems[10].

Here we take a similar approach to climate models to validate our model internally (that is, within our dataset). Climate models can be validated by allowing them to condition on data and then run freely for some period. If the model successfully retrodicts conditions at a future point in time, it provides evidence that the model captures the dynamics of interest. We follow much the same approach here, simulating total engagement from the initial tweet throughout an event. At the coarsest level, the total number of observed posts (10.4 million) falls within the 89% CI of our baseline simulations (10.8 million, 89% CI, (9.8, 11.7)). On the scale of individual events, posterior predictive simulations recover the number of observed posts over several orders of magnitude, despite the model only being seeded with posts in the first time step and the time-varying empirical follower distribution (Supplementary Fig. 2). This holds true across several orders of magnitude in post volume and for events that vary widely in duration from one hour to several days. Visual inspection of posterior-predictive time series similarly indicates that our model recovers fine-grained temporal dynamics, even for our largest events where the number of data points far exceeds the model parameters (Supplementary Fig. 3). Considering the relatively small number of parameters (five in this model), this provides evidence that our model is adequately capturing key features of the underlying temporal dynamics.

**Post-event engagement.** Our model cannot directly evaluate post-event engagement, as it is designed to capture viral spreading dynamics rather than long, noisy periods of posting about a topic. These periods would be difficult to capture directly with a generative model, making it challenging to infer the impact of interventions on misinformation about a topic in general. However, there exists a quite regular relationship between the proportion of posts that occur within our definition of an event and those that occur subsequent to the event (Fig. 4c).

We can leverage this fact to gain insight into how interventions may impact discussion following the viral periods we analysed. To accomplish this, we used a Bayesian log-normal regression to estimate the effect of posts within the largest event on subsequent engagement (Supplementary Table 10):

$$\beta \sim \text{Cauchy}(0, 1)$$
$$\sigma \sim \text{Cauchy}(0, 1)$$
$$\mu = \beta x$$
$$y \sim \text{LogNormal}(\mu, \sigma)$$

Here, $y$ is post-event engagement, and $x$ is engagement during the largest event. The intercept is set at zero, as an event with no posts would not be expected to produce subsequent posts. We then use the posterior distribution from this model to estimate subsequent engagement as a function of engagement during our simulated events with intervention. This is summed across events to generate the estimates shown in Fig. 4d. This method provides insight, but we note that it is limited by the assumption that the relationship between within- and post-event engagement is invariant to interventions. Furthermore, it is limited by the extent to which our data collection process captured posts across the entire incident (that is, event and subsequent posts).

**Reporting summary.** Further information on research design is available in the Nature Research Reporting Summary linked to this article.

## Data availability
Given Twitter's data use agreement, we cannot release the full dataset. However, we have made available aggregated time series sufficient to reproduce our findings. The data to reproduce the results are available on Zenodo (https://doi.org/10.5281/zenodo.6480218).

## Code availability
The code to reproduce the results is available on Zenodo (https://doi.org/10.5281/zenodo.6478446). Any updates to the code can be found on GitHub (https://github.com/josephbb/CombinedPoliciesMisinfo).

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

## Acknowledgements

This work was made possible through support from the John S. and James L. Knight Foundation, the UW Center for an Informed Public, the University of Washington eScience Institute and Craig Newmark Philanthropies. J.D.W., E.S.S. and K.S. acknowledge the support of the National Science Foundation (award no. 2027792). K.S. acknowledges the support of the National Science Foundation (NSF CAREER award no. 1749815). The funders had no role in study design, data collection and analysis, decision to publish or preparation of the manuscript. We also thank our collaborators, the Stanford Internet Observatory, Graphika, DFRLab and the Electoral Integrity Partnership. We further thank C. Bergstrom, I. Couzin, F. Rossine, R. Moran and K. Koltai for their feedback.

## Author contributions

J.B.B.-C., A.B. and J.D.W. conceived the study. J.S.S., A.B., M.W., I.K., E.S.S. and K.S. developed the dataset. J.B.B.-C. and J.D.W. wrote the model and simulation code. J.B.B.-C. drafted the initial manuscript, and all authors were involved in subsequent revisions.

## Competing interests

The authors declare no competing interests.

## Additional information

**Correspondence and requests for materials** should be addressed to Joseph B. Bak-Coleman.

# Reporting Summary

## Statistics

For all statistical analyses, confirm that the following items are present in the figure legend, table legend, main text, or Methods section.

| n/a | Confirmed | |
|---|---|---|
| ☒ | ☐ | The exact sample size (*n*) for each experimental group/condition, given as a discrete number and unit of measurement |
| ☒ | ☐ | A statement on whether measurements were taken from distinct samples or whether the same sample was measured repeatedly |
| ☒ | ☐ | The statistical test(s) used AND whether they are one- or two-sided *Only common tests should be described solely by name; describe more complex techniques in the Methods section.* |
| ☐ | ☒ | A description of all covariates tested |
| ☐ | ☒ | A description of any assumptions or corrections, such as tests of normality and adjustment for multiple comparisons |
| ☐ | ☒ | A full description of the statistical parameters including central tendency (e.g. means) or other basic estimates (e.g. regression coefficient) AND variation (e.g. standard deviation) or associated estimates of uncertainty (e.g. confidence intervals) |
| ☒ | ☐ | For null hypothesis testing, the test statistic (e.g. *F*, *t*, *r*) with confidence intervals, effect sizes, degrees of freedom and *P* value noted *Give P values as exact values whenever suitable.* |
| ☐ | ☒ | For Bayesian analysis, information on the choice of priors and Markov chain Monte Carlo settings |
| ☐ | ☒ | For hierarchical and complex designs, identification of the appropriate level for tests and full reporting of outcomes |
| ☐ | ☒ | Estimates of effect sizes (e.g. Cohen's *d*, Pearson's *r*), indicating how they were calculated |

*Our web collection on statistics for biologists contains articles on many of the points above.*

## Software and code

Policy information about availability of computer code

| Data collection | Data were collected using the Twitter API in real time during the 2020 US presidential Election. Collection began on 1 September 2020 and ran through Dec 15th, 2020. Collection involved an evolving set of keywords in response to emerging narratives regarding electoral misinformation identified by the Electoral Integrity Partnership. |
|---|---|
| Data analysis | All data analysis was conducting in Python (Python 3.9.10) leveraging the standard Scipy stack and Pystan (v. 2.19.1.1). Versions of all installed python packages are available in the "requirements.txt" file of the code respos[i]tory. Code to reproduce the results is available on Zenodo (https://doi.org/10.5281/zenodo.6478446). Any updates to the code can be found on Github (https://github.com/josephbb/ CombinedPoliciesMisinfo) |

For manuscripts utilizing custom algorithms or software that are central to the research but not yet described in published literature, software must be made available to editors and reviewers. We strongly encourage code deposition in a community repository (e.g. GitHub). See the Nature Portfolio guidelines for submitting code & software for further information.

## Data

Policy information about availability of data

All manuscripts must include a data availability statement. This statement should provide the following information, where applicable:

- Accession codes, unique identifiers, or web links for publicly available datasets
- A description of any restrictions on data availability
- For clinical datasets or third party data, please ensure that the statement adheres to our policy

Given Twitter's Data Use-Agreement, we cannot release the full dataset. However, we have made available aggregated time-series sufficient to reproduce our findings. Data to reproduce the results are available on Zenodo (https://doi.org/10.5281/zenodo.6480218).

# Field-specific reporting

Please select the one below that is the best fit for your research. If you are not sure, read the appropriate sections before making your selection.

☐ Life sciences ☒ Behavioural & social sciences ☐ Ecological, evolutionary & environmental sciences

For a reference copy of the document with all sections, see nature.com/documents/nr-reporting-summary-flat.pdf

# Behavioural & social sciences study design

All studies must disclose on these points even when the disclosure is negative.

| | |
|---|---|
| Study description | Our study involved deriving a model of viral information spread to examine the impact of interventions aimed at reducing the spread of viral misinformation. To do so, we collected a large number of Tweets related to the US 2020 Presidential election. Mixed-methods approaches were used to identify distinct incidents (specific narratives) of misinformation and associated search terms. These search terms were used to distinguish incidents and extract relevant tweets from our larger database. We then used quantitative methods to segment incidents into distinct events. Finally, we used this dataset to derive a mathematical model of misinformation spread and examine the likely impact of commonly proposed interventions. |
| Research sample | Our sample involves users on Twitter that posted about topics related to the 2020 US presidential elections. Given the large initial volume of Tweets collected and the large team of individuals used to identify and collect misinformation narratives, we believe the dataset is representative of English-language tweets in the US about the 2020 presidential election. This sample was used as the electoral integrity partnership and Twitter's open API provided a unique opportunity to gather the data needed to apply our approach. |
| Sampling strategy | We sampled randomly and used all tweets that matched the criteria. A sample size calculation was not necessary. |
| Data collection | All data were collected in accordance with the University of Washington Institutional Review Board. Our dataset was collected in real-time during the 2020 US election. We relied on a set of 160 keywords to collect posts from Twitter's API (1.04 billion). Keywords were updated in response to new narratives, for instance, adding "sharpiegate" and related terms after false narratives emerged about the use of Sharpie markers invalidating ballots. Working with the Electoral Integrity Partnership, we cataloged instances of false or misleading narratives that were either detected by the team or reported by external partners.. This led to a large corpus of tickets associated with validated reports of misleading, viral information about election integrity.<br><br>Tickets that shared a common theme were consolidated into incidents. We developed search terms and a relevant date range for each incident to query posts from our tweet database. Incidents (N=430) were generally characterized by one or more periods of intense activity followed by returning to a baseline state. Search terms and descriptions of incidents are provided along with the data. |
| Timing | September 1st 2020 to December 15th 2020 |
| Data exclusions | Our model was fit to each event using PyStan 2.9.1.1. We fit events separately (rather than hierarchically) as they varied widely in their time scales, magnitudes, and context within the broader 2020 election cycle. Of the 544 candidate events, our model performed well on 454 events (10.4 Million posts/tweets) of rapid misinformation spread. Our model was unlikely to be suitable for all events as it makes the assumption that post volume is well predicted by the number of previously exposed accounts on Twitter.<br><br>To safeguard against this, we relied on a number of criteria to ensure model fit to a given event. Events were included in the final analysis if a) the posterior 89% C.I. of total posts contained the observed value and b) the chains successfully converged for all parameters ($r\_hat < 1.1$) c) The fit did not contain divergent transitions and d) the event lasted longer than an hour (i.e., >12 data points to fit). Other than these criteria, events surrounding the Dominion narrative were removed as they involved long periods high volume online discussion. This filtering process resulted in the inclusion of 454 events (83% of total events), and 10.4 million posts. |
| Non-participation | Analyses were conducted on digital trace data, and no participants were explicitly used. |

| Randomization | This study does not have explicit experimental and control groups, so no randomization was needed. |
| --- | --- |

# Reporting for specific materials, systems and methods

We require information from authors about some types of materials, experimental systems and methods used in many studies. Here, indicate whether each material, system or method listed is relevant to your study. If you are not sure if a list item applies to your research, read the appropriate section before selecting a response.

## Materials & experimental systems

| n/a | Involved in the study |
| --- | --- |
| ☒ | ☐ Antibodies |
| ☒ | ☐ Eukaryotic cell lines |
| ☒ | ☐ Palaeontology and archaeology |
| ☒ | ☐ Animals and other organisms |
| ☐ | ☒ Human research participants |
| ☒ | ☐ Clinical data |
| ☒ | ☐ Dual use research of concern |

## Methods

| n/a | Involved in the study |
| --- | --- |
| ☒ | ☐ ChIP-seq |
| ☒ | ☐ Flow cytometry |
| ☒ | ☐ MRI-based neuroimaging |

## Human research participants

Policy information about studies involving human research participants

| Population characteristics | Users on Twitter |
| --- | --- |
| Recruitment | Data were collected via the API as they were publicly available. |
| Ethics oversight | University of Washington IRB |

Note that full information on the approval of the study protocol must also be provided in the manuscript.

