## [Peer Review File · Nature Human Behaviour]

Peer Review Information

Journal: Nature Human Behaviour

Manuscript Title: Combining interventions to reduce the spread of viral misinformation

Corresponding author name(s): Joseph B. Bak-Coleman

Reviewer Comments & Decisions:

Decision Letter, initial version:

15th November 2021

Dear Dr Bak-Coleman,

Thank you once again for your manuscript, entitled "Combining interventions to reduce the spread of viral misinformation", and for your patience during the peer review process.

Your Article has now been evaluated by 3 referees. You will see from their comments copied below that, although they find your work of potential interest, they have raised quite substantial concerns. In light of these comments, we cannot accept the manuscript for publication, but would be interested in considering a revised version if you are willing and able to fully address reviewer and editorial concerns.

We hope you will find the referees' comments useful as you decide how to proceed. If you wish to submit a substantially revised manuscript, please bear in mind that we will be reluctant to approach the referees again in the absence of major revisions. We are committed to providing a fair and constructive peer-review process. Do not hesitate to contact us if there are specific requests from the reviewers that you believe are technically impossible or unlikely to yield a meaningful outcome.

In your revision, we ask you to fully address all of the reviewers' concerns. In addressing the issues raised by Reviewers #1 and #3 and related to your modelling assumptions and choices (the mode of viral spread as a simple contagion, not accounting for network topology, your operationalization of virality, and others), please ensure that they are realistic and take into account what is known about information spreading phenomena. Please include new and additional modelling and analyses, as needed.

Reviewer #2 notes that, while your model is calibrated using data, there is no real-world validation of your proposed interventions. While we understand that this may not be possible given the limitations of your data, we ask you to provide a discussion of whether your insights would generalize if implemented in the real-world setting. Provide evidence only if this is possible or feasible to do.

Reviewer #3 cites a paper by Qiu et al (Nature Human Behaviour, 1(7), 1-7). We note that this article has been retracted due to code and analysis errors, and we ask you to not cite it in your work.

If you wish to submit a suitably revised manuscript we would hope to receive it within 6 months. We understand that the COVID-19 pandemic is causing significant disruptions which may prevent you from carrying out the additional work required for resubmission of your manuscript within this timeframe. If you are unable to submit your revised manuscript within 6 months, please let us know. We will be happy to extend the submission date to enable you to complete your work on the revision.

- Include a "Response to the editors and reviewers" document detailing, point-by-point, how you addressed each editor and referee comment. If no action was taken to address a point, you must provide a compelling argument. This response will be used by the editors to evaluate your revision and sent back to the reviewers along with the revised manuscript.
- Highlight all changes made to your manuscript or provide us with a version that tracks changes.

[REDACTED]

This URL links to your confidential home page and associated information about manuscripts you may have submitted, or that you are reviewing for us. If you wish to forward this email to co-authors, please delete the link to your homepage.

Thank you for the opportunity to review your work. Please do not hesitate to contact me if you have any questions or would like to discuss the required revisions further.

Sincerely,

Arunas Radzvilavicius, PhD
Editor
Nature Human Behaviour

Reviewer expertise:

Reviewer #1: computational social science, misinformation

Reviewer #2: network spreading models, CSS, misinformation

Reviewer #3: network science, spreading models

REVIEWER COMMENTS:

Reviewer #1:
Remarks to the Author:

This paper addresses how social media platforms could use specific policies to reduce the spread of fake news. Using real data on engagement with fake news and a Bayesian model that can simulate the effects of various policies, they estimate how much these policies would have reduced tweets about fake news in 2020 under different implementation scenarios, with policies considered both independently and in combination. The main finding is that no individual proposed policy would be effective under realistic implementation settings, but that combining these policies (using realistically feasible settings) would cause useful reductions.

The results seem reasonable. Of course, readers could debate the details of what thresholds make a policy realistic or effective, or how correctly the model simulates the effect of (e.g.,) banning or nudging users. But the real significance of this work, in my opinion, is that it offers an empirical (data-backed) framework by which we can start to compare and evaluate proposed solutions to the problem of fake news. In particular, the project goes all the way from data collection and representation choices, through model development and simulation, to discussing policies from the points of view of the platforms and users. Such a large scope is rare. I read both the modeling and the policy components of this work as solid and sophisticated; and certainly the topic of fake news continues to be timely and of interest to a wide audience.

My biggest concern is the relative lack of discussion of the data, its representativeness, its properties, and how these properties (or other principles) guided the development of this particular model. I found it difficult to understand the details and limitations of the data and model from the main text. When I examined the replication package, it made much more sense—but brought new questions to mind. (Note: I ran about half the code, up through the model fitting, which was enough to convince me of the rest.) Since the model is a key contribution, I think ideally more material from the Methods section might be woven in earlier, in order to avoid surprises and make the paper easier to read in a single pass.

Some high-level questions & comments on those topics.
Data & collection:

-If keyword searches were developed in real time, does that mean we're missing posts from before the researchers started tracking a story—or did the search API include history?

-Were these fake news stories likely the biggest? What can we say about the fake news we're missing?

-It seems like a big oversight to neglect to mention the number of users in the data set (for the account banning analysis), the precise time period covered, and the structure of the input data (namely, it's the number of posts in each 5-min span, and for each post, the account's number of followers). (It's worth emphasizing that y_t represents the number of posts, not the number of distinct users--because we should expect large variation in how often users post.)

-Do temporal patterns of engagement around fake news stories look similar to those for non-fake news (or does that even matter)? What does the literature tell us that these look like? (I'm less familiar with branching processes and SEIR models, but expected some discussion of the many existing models of online information spread, for context.)

Model:

-Briefly, what makes the modeling (a) challenging, and (b) solvable? (The next two points are trying to get at this in different ways.)

-It's clear that both the data segmentation and the model development are premised on particular patterns -- e.g., exactly one large spike, possibly(?) early on in an event -- but these (and their rationale) did not come through for me. (Or maybe I'm misguided, since there's also the phrase "given the wide range of possible event shapes.")

-Not "why is it ok to use a simple model?", but rather, "why did we choose to have the modeled quantities be ..." [a story's baseline rate of discussion; the log of the number of followers exposed just now; and a momentum-like term, called virality, that incorporates the past number of people exposed, but decays over time]. Perfectly reasonable answers might include "it's been shown to work before," "it's computationally feasible," and/or "these quantities make it straightforward to simulate the interventions we want."

-Is it even appropriate to call it "a [single] model," when (as I was surprised to discover) the parameters are not tied across events? "A model estimated from 6M posts" feels a lot different than "a separate version of the model for each of 216 events, where each event has at least 12 data points." Seems like (possibly) a lot of free parameters.

-Framing: the approach makes sense when I think of it as a way of using empirical data to simulate new similar data (such as in bootstrapping)—but that's not what the phrase "generative model" had me anticipating.

-Is it reasonable to have to discard 20% of events, since the model cannot fit them?

-Any thoughts on coordinated misinformation campaigns and (e.g.) whether their dynamics would violate core assumptions of the model?

Lower-level questions, inconsistencies, unclear parts.

-In Methods, unclear numbers: $n = 153$ or 154 , events = 260 or 220 or 216 , simulations = 500 or 100 .

-The code vs. the writeup disagree on $\delta \sim \text{beta}(1,1)$ vs. $\text{beta}(2, 2)$, and on ϕ and λ as exponential vs. half-exponential.

-Add 'output/figures/Events' to 'create_output_directories()'

-Initially unclear that: "engagement" = "post" = number of any type of post, on the fake news items they've collected, "cumulative engagement" in y-axis refers to the number of tweets posted (not, e.g., users).

-Fig 1D: why do the axes differ from 1C?

-(Account Banning) Should we expect verified accounts to be more, or less, important to fake news than unverified accounts? Do they have more followers, generally?

-The two paragraphs after Fig 4: is this looking at posts outside of all events, or posts inside events but outside the largest event? (Typo: the text's values don't match the SI.)

Reviewer #2:

Remarks to the Author:

Dear editor,

The work submitted by Bak-Coleman et al. for consideration in Nature Human Behavior addresses an interesting and timely question: how can social media platforms reduce the spread of misinformation without having to resort to mass censorship? The authors employ a computational statistics approach to show that even though several of the interventions proposed so far in the literature have little impact when deployed in isolation from each other, their combination leads to important synergies, achieving in one case a reduction of ~50% of the circulating misinformation.

This seems like a remarkable result, especially in light of the recent revelations in the press about Facebook. The work could have broad impact, and would likely be of interest to an interdisciplinary audience.

There are, however, a number of issues with the manuscript, and I am unable to recommend acceptance of the manuscript as is. Before I can recommend publication in Nature Human Behavior, I hope the authors would consider addressing the following issues:

- To prove their main point, the authors extrapolate from a statistical model without any form of empirical validation. The fact that the model is well calibrated to the original data (without interventions) does not tell us anything about the more important question of whether the same model (with the various interventions) would be a "good" model to describe the same process under those interventions. In this sense, the type of evidence the authors can achieve is purely computational but not causal. I am wondering whether there are ways to provide more causal evidence in support of the main findings (i.e. that combining interventions leads to large reductions in spread), but lacking data about the interventions deployed by the platform (in this case Twitter), it is hard to see how one could go beyond mere speculation guided by a model (albeit a well calibrated one).
- Related to the first point, the authors implicitly assume that the data they observe correspond to the scenario in which the platform is not making any intervention, however it is quite likely that Twitter may be already applying some of the interventions studied. In this case, the observed reductions may be biased toward overestimating the amount of reduction one could achieve.
- The framing of the results is a bit over-optimistic. True, the findings show that there are alternative methods to extreme censorship (something no platform would consider anyway, at least in the US), however a more pessimistic interpretation of the findings could be that, in order to be effective, interventions will have to necessarily include many different moving parts, making them more complex compared to an approach based on a single intervention.

Some other minor issues:

- A couple of passages in the methods section are a bit unclear. In particular I did not understand what is the baseline rate of detection for related keywords (line 490) and what is the reason for using linear interpolation for normalization (line 547).
- The plots in the supplementary information are too small to be legible. Reducing the number of plots per page could help.
- There are few typos in the manuscript (I spotted some on line 12 (Capitol instead of capital) and 13 (online ... online)), and some sentences were a bit hard to parse (e.g. line 402). I would recommend some more editing especially in the Methods section.

- The plots are not very well readable in B&W. This is especially the case of Fig 1D, 2, 3, and 4.

Reviewer #3:

Remarks to the Author:

After multiple readings of this paper I suggest a rejection for Nature Human Behaviour, for the reasons I will list in the following lines. Nevertheless I would like to encourage the authors to add some other huge analysis to their work, I will provide some hints.

The authors present a new model to measure misinformation spreading and then they consider different policies to reduce misinformation, implementing them in their own model and comparing the results.

First of all, the model lay on highly questionable assumptions. The authors assume that the misinformation spreading follows a "simple contagion" dynamics: this should be better explained and discussed since several works in literature show the opposite thesis. Some references:

- Centola D, Macy M (2007) Complex contagions and the weakness of long ties. *Am J Sociol* 113(3):702–734

- Lerman K (2016) Information is not a virus, and other consequences of human cognitive limits. *Future Internet* 8(2):21

- Romero DM, Meeder B, Kleinberg J (2011) Differences in the mechanics of information diffusion across topics: idioms, political hashtags, and complex contagion on twitter. In: *Proceedings of the 20th International Conference on World Wide Web*. ACM. pp 695–704

- Min B, San Miguel M (2018) Competing contagion processes: Complex contagion triggered by simple contagion. *Sci Rep* 8(1):10422

Second, the users' audience is approximated by their follower count, and this is basically what defines the virality (with the addition of a decay). In my opinion here the authors should better elaborate what is their definition of virality and what they are measuring: virality is something related to influence and influence is a very complex concept and does not necessarily correlates with the number of followers. In other words, the authors propose a model in which a high number of followers lead to high virality, but there are several famous paper that show that this is not what actually happen in social media, or that other factors as limited user attention are very important. Some references:

- Cha, M., Haddadi, H., Benevenuto, F., & Gummadi, K. (2010, May). Measuring user influence in twitter: The million follower fallacy. In *Proceedings of the international AAAI conference on web and social media* (Vol. 4, No. 1).

- Qiu, X., Oliveira, D. F., Shirazi, A. S., Flammini, A., & Menczer, F. (2017). Limited individual attention and online virality of low-quality information. *Nature Human Behaviour*, 1(7), 1-7.

Finally, the model does not take into account the network itself, its topology, that actually plays a very crucial role in spreading phenomena.

Second, the authors used a dataset to tune the parameters, and then, using the obtained values, they simulated several interventions. Nevertheless, a bit of analytical approach could have been useful to better understand the role of the parameters: I suggest some heatmaps varying two parameters on

x, y and measuring in $z(x_i, y_i)$ the averaged engagement over several simulations with those values $x=x_i$ and $y=y_i$.

Then the model could be better understood and maybe compared with the existing ones: about this point, I would also suggest to add a comparison with other existing models and a simple SIR, and proving that this model capture the dynamics better than other models.

These are the main reasons that made me lean towards rejections. As I mentioned before, in my opinion there are many problems related to the model itself, while the rest of the paper and methodology used to test several interventions are quite ok. It could be interesting to study these interventions also in other proposed models.

Minor issues:

In the model equation, it is not really necessary to define $x(t)$ separately since it can create a bit of confusion.

Author Rebuttal to Initial comments

Reponse to Reviewers: Combining interventions to reduce the spread of viral misinformation online.

REVIEWER COMMENTS:

Reviewer #1:

Remarks to the Author:

R1: This paper addresses how social media platforms could use specific policies to reduce the spread of fake news. Using real data on engagement with fake news and a Bayesian model that can simulate the effects of various policies, they estimate how much these policies would have reduced tweets about fake news in 2020 under different implementation scenarios, with policies considered both independently and in combination. The main finding is that no individual proposed policy would be effective under realistic implementation settings, but that combining these policies (using realistically feasible settings) would cause useful reductions.

The results seem reasonable. Of course, readers could debate the details of what thresholds make a policy realistic or effective, or how correctly the model simulates the effect of (e.g.,) banning or nudging users. But the real significance of this work, in my opinion, is that it offers an empirical (data-backed) framework by which we can start to compare and evaluate proposed solutions to the problem of fake news. In particular, the project goes all the way from data collection and

representation choices, through model development and simulation, to discussing policies from the points of view of the platforms and users. Such a large scope is rare. I read both the modeling and the policy components of this work as solid and sophisticated; and certainly the topic of fake news continues to be timely and of interest to a wide audience.

My biggest concern is the relative lack of discussion of the data, its representativeness, its properties, and how these properties (or other principles) guided the development of this particular model. I found it difficult to understand the details and limitations of the data and model from the main text. When I examined the replication package, it made much more sense—but brought new questions to mind. (Note: I ran about half the code, up through the model fitting, which was enough to convince me of the rest.) Since the model is a key contribution, I think ideally more material from the Methods section might be woven in earlier, in order to avoid surprises and make the paper easier to read in a single pass.

AR: The reviewer's biggest concern is the lack of details concerning the data used to test the model. This is a reasonable critique and we agree with the reviewer that we need to better address this in the paper. We have added several new paragraphs that address this, at the outset of the results section and in the methods. Further, we re-organized the presentation of methods and results to better orient the reader and comply with article guidelines.

More specifically, we begin the results section with a “Data and Model Overview” section that provides a high-level summary of our data collection process and model fitting/simulation procedure. As the reviewer suggests, we hope this orients the reader sufficiently to allow them to follow the results more critically. As word limits preclude a very comprehensive discussion in the results, we further expand upon the data collection in the methods in a section entitled “Data collection and processing”. To further accommodate reviewer suggestions and word limits, we’ve moved much of the math into the methods. In the process of integrating the math more thoroughly into the methods, we’ve greatly revised that section in a way that we believe will clarify many points of concerns for the readers and raised by the reviewers.

R1: Some high-level questions & comments on those topics. Data & collection:

-If keyword searches were developed in real time, does that mean we're missing posts from before the researchers started tracking a story—or did the search API include history?

AR: Because the broader election-related dataset was continuously collected (i.e., real time without history), we have access to all of the tweets that matched the 160 generic

election-related keywords (though some keywords were added later in the study period). Though it's possible that we miss some early tweets connected to a misinformation story, we

visually examined the growth of each of our stories to make sure that we didn't exclude early important tweets. Smooth growth early on is an indicator that our collection did not begin halfway through a viral event.

R1: -Were these fake news stories likely the biggest? What can we say about the fake news we're missing?

AR: *We tracked 8 stories that had more than 1 million tweets connected to them (not necessarily in the time period evaluated as a viral event). There were dozens more that had more than 100k associated tweets. As our collection arose from observation by dozens of researchers working full-time, we do not believe we missed any of the larger narratives throughout the course of the election. However, some of the larger narratives may not have been well fit by our model. Figure S2 highlights the sizes of all events. While it's possible that we missed smaller stories, we think it's very unlikely that there was a story as large as these 8 that we missed, but we are open to suggestions that we can look further into. Moreover, improving our dataset and nearly doubling its size did not alter our key findings, which suggests some robustness to missing tweets.*

R1: -It seems like a big oversight to neglect to mention the number of users in the data set (for the account banning analysis), the precise time period covered, and the structure of the input data (namely, it's the number of posts in each 5-min span, and for each post, the account's number of followers). (It's worth emphasizing that y_t represents the number of posts, not the number of distinct users--because we should expect large variation in how often users post.)

AR: *We have incorporated the requested information/emphasis throughout, where appropriate.*

R1: -Do temporal patterns of engagement around fake news stories look similar to those for non-fake news (or does that even matter)? What does the literature tell us that these look like? (I'm less familiar with branching processes and SEIR models, but expected some discussion of the many existing models of online information spread, for context.)

AR: *We didn't explicitly contrast false and true news stories in our analysis. However, based on a recent paper (below), fake news stories and non-fake news stories follow similar dynamics.*

Notably, this paper also relied on simple contagion models.

Juul, J. L., & Ugander, J. (2021). Comparing information diffusion mechanisms by matching on cascade size. *Proceedings of the National Academy of Sciences of the United States of America*, 118(46). <https://doi.org/10.1073/pnas.2100786118>

R1: Model:

-Briefly, what makes the modeling (a) challenging, and (b) solvable? (The next two points are trying to get at this in different ways.)

AR: *This is a difficult modeling problem because we have very limited insight into the algorithmic choices that decide who sees what content. As a consequence, it is impossible to know the “network” of interactions as they exist. A user may only see a (non-random) fraction of those they follow and this can only be inferred indirectly by making tenuous assumptions about what constitutes a network tie. Common approaches to modeling information spread that invoke a given definition of an edge and subsequent network topology will yield results that can be biased by the choices used to construct that topology. Even if we had an agreeable definition of an edge, rate limits and the data volume inherent to an entire election would undermine feasibility of constructing that network for a large event, much less reasonably constructing it dynamically throughout the course of the election and across events.*

Ultimately for a network-based simulation, it would be unclear if model fit indicated the dynamics, or the dynamics biased by the assumptions underlying the definition of the network. These are non-trivial challenges, although not too distinct from those faced by modeling of infectious disease. Our daily interactions and air-flow patterns create remarkably complex and hopelessly difficult to model topologies of disease spread. Yet, failing to model the likely efficacy of interventions (in search of a more perfect model) abdicates a fair deal of urgently needed scientific oversight. We have incorporated some of this discussion throughout.

What allows us to make inferential progress is in accepting that a model of a simple contagion will be informative (if not perfectly correct). A key feature of a simple contagion is that it requires a minimum of one interaction for an individual to adopt the state. We know this to be true on Twitter (only one person needs to share something for it to find its way into a feed). We also know from other lines of evidence (e.g., Martin 2016), that engagement is a nearly linear function of user follower count. Out-degree being linear with transmission is something we would expect to see with a simple contagion. The same relationship would not be observed with common varieties of complex contagions (i.e. more linked to the clustering coefficient of a network or sub-community within that network, etc..).

Our assumption of a simple contagion—in conjunction with follower counts spanning multiple orders of magnitude—our model manages to capture dominant features of the empirical dynamics driven by something mechanistically simple and similar to disease super-spreaders. The correspondence between simulated and observed effect sizes in Figure 2 suggests that our model is a useful inferential tool.

Martin, T., Hofman, J. M., Sharma, A., Anderson, A., & Watts, D. J. (2016). Exploring limits to prediction in complex social systems. *25th International World Wide Web Conference, WWW 2016*, 683–694. <https://doi.org/10.1145/2872427.2883001>

R1: -It's clear that both the data segmentation and the model development are premised on particular patterns -- e.g., exactly one large spike, possibly(?) early on in an event -- but these(and their rationale) did not come through for me. (Or maybe I'm misguided, since there's also the phrase "given the wide range of possible event shapes.")

AR: *We believe this stems from a lack of clarity in our write-up and have improved our description of segmentation and rationale.*

"We then wished to extract segments of the timeseries that exhibit macroscopic features consistent with viral dynamics. More specifically, candidate events should be comprised of quiescent periods before and after the event where our search terms return to baseline levels. However, multiple peaks may occur between these boundaries. To extract candidate events, we computed the raw timeseries of post volume per five minutes for each of our distinct incidents."

R1: -Not "why is it ok to use a simple model?", but rather, "why did we choose to have the modeled quantities be ..." [a story's baseline rate of discussion; the log of the number of followers exposed just now; and a momentum-like term, called virality, that incorporates the past number of people exposed, but decays over time]. Perfectly reasonable answers might include "it's been shown to work before," "it's computationally feasible," and/or "these quantities make it straightforward to simulate the interventions we want."

AR: *The reviewer has certainly anticipated our rationale, more or less. We note that the log number of followers is only used in the linear approximation and is passed through an exponential to compute the expected value. In other words, we're assuming that spread is linearly proportional to those exposed. This has been shown to work before in countless models of disease (notably Lloyd-Smith 2005, which inspired our paper) and dynamics of false information spread online (most recently Juul et al 2021). Disease models also provide quite a bit of insight even in the absence of explicit network structure which is not strictly known for Twitter, due to algorithmic filtering. However, incorporating variation in individuals' ability to transmit is critical (As per Lloyd-Smith 2005). Moreover, some form of decay felt necessary based on the observation that newer actions appear more likely to be seen in ones' feed. At the end of the day, this model felt the most justified by work in other domains, while being computationally feasible, possible given our data, and is the simplest model that would allow us to compare commonly proposed interventions. A simple SIR (SEIR) model would not allow us to evaluate specific account removal across all events, for example.*

Similarly, a model of complex contagion would not be possible without adding considerable epistemic uncertainty in defining the structure of the network and (perhaps novel) mechanism of contagion.

R1: -Is it even appropriate to call it "a [single] model," when (as I was surprised to discover) the parameters are not tied across events? "A model estimated from 6M posts" feels a lot different than "a separate version of the model for each of 216 events, where each event has at least 12 data points." Seems like (possibly) a lot of free parameters.

AR: *Our language here follows precedent in the statistical literature (McElreath 2020) whereby the model typically refers to the choice of likelihood and priors. In developing this paper we considered whether some form of parameter pooling (i.e, fitting all events simultaneously and modeling parameters within and across events) was a) possible and b) advisable. Pooling could reduce the number of effective parameters and help manage under/overfit tradeoffs across the dataset.*

The simplest approach to doing so would be to assume the events are exchangeable yet drawn from some common distribution. This seems unreasonable given qualitative differences between the types of content and myriad ways in which user behavior is likely to change over the course of a disinformation campaign. That leaves us needing to model how the parameters change over the course of the election, which is very uncharted territory. We also suspect unmeasured mediation in parameters owing to general activity on Twitter and competing stories would make pooling fraught or difficult. Computational tractability aside, it's hard to coherently think of a model that would allow us to estimate the joint probability across events in a way that we could be sure isn't biasing our results. Moreover, such a model would miss events that are not captured by viral dynamics but nonetheless impact parameters, which we suspect would be a major source of bias in the resulting estimations.

R1: -Framing: the approach makes sense when I think of it as a way of using empirical data to simulate new similar data (such as in bootstrapping)—but that's not what the phrase "generative model" had me anticipating.

AR: *Our use of generative terminology from the Bayesian inference literature wherein we seek to estimate the joint probability distribution of the data and parameters given a stated set of relations between the two, namely our model (McElreath 2020). This would stand in contrast to estimating the marginal probability of the data conditioned on a model with fixed parameters (i.e. hypothesis testing). We do empirically draw from the follower distributions (similar to bootstrapping) although this is more of a work-around to allow us to remove users from the distribution which would not be possible if we fit it as a Pareto distribution or whatnot. We note that we do not sample from the true numbers of posts that are generated by the joint probability distribution of the parameters and the other observables (i.e. initial post volume and distribution of account sizes).*

R1: -Is it reasonable to have to discard 20% of events, since the model cannot fit them?

AR: *We doubt that any single formulation of a model could capture the totality of dynamics of misinformation spread, particularly when a narrative is predominantly driven by off-platform dynamics (e.g. news, coordination, overflow from other social media sites). Given this prior knowledge, it would have been alarming (to us) if our model happily fit every shape thrown at it. In designing the study, we relied on strict inclusion criteria (Posterior predictive checks, HMC diagnostics) to reduce epistemic uncertainty and ensure that we were only including events that we could be reasonably certain are well-approximated by our model. This motivates our decision to limit our discussion and write-up to viral misinformation.*

R1: -Any thoughts on coordinated misinformation campaigns and (e.g.) whether their dynamics would violate core assumptions of the model?

AR: To some extent, our simulations are implicitly conditioned on coordination. Briefly, if a network of large accounts tends to share early in a cascade, this is incorporated in the structure of simulations. When our simulations draw from follower account size distributions at a given time-step, coordination will make them more likely to show up early (when influence is maximal). In this sense, coordination by a smaller fraction of (even quite large) accounts should not dramatically impact the findings. However, we cannot rule out that certain approaches to coordination would impact the efficacy of interventions. In these cases, account removal may be more effective than anticipated. We hope that future work can build out in this direction

R1: Lower-level questions, inconsistencies, unclear parts.

-In Methods, unclear numbers: $n = 153$ or 154 , events = 260 or 220 or 216, simulations = 500 or 100.

AR: *We have clarified throughout and updated to be consistent with our larger dataset.*

R1: -The code vs. the writeup disagree on $\delta \sim \text{beta}(1,1)$ vs. $\text{beta}(2, 2)$, and on ϕ and λ as exponential vs. half-exponential.

AR: *Fixed for the beta prior to (1, 1). We note that for phi and lambda the <lower=0> constraint in Stan treats the distribution as half-exponential.*

-Add 'output/figures/Events' to 'create_output_directories()'

AR: *Done*

R1: -Initially unclear that: "engagement" = "post" = number of any type of post, on the fake news items they've collected, "cumulative engagement" in y-axes refers to the number of tweets posted (not, e.g., users).

AR: *We simplified our language and clarified this throughout.*

R1: -Fig 1D: why do the axes differ from 1C?

AR: *Fixed.*

R1: -(Account Banning) Should we expect verified accounts to be more, or less, important to fake news than unverified accounts? Do they have more followers, generally?

AR: *They do tend to have more followers in general, but (internally) we have not been able to identify any increase in their engagement beyond # of followers. It is a bit difficult to disentangle because the probability of being verified increases with follower count, so we haven't published anything to this effect.*

R1: -The two paragraphs after Fig 4: is this looking at posts outside of all events, or posts inside events but outside the largest event? (Typo: the text's values don't match the SI.)

Reviewer #2:

Remarks to the Author:

Dear editor,

R2: The work submitted by Bak-Coleman et al. for consideration in Nature Human Behavior addresses an interesting and timely question: how can social media platforms reduce the spread of misinformation without having to resort to mass censorship? The authors employ a computational statistics approach to show that even though several of the interventions proposed so far in the literature have little impact when deployed in isolation from each other, their combination leads to important synergies, achieving in one case a reduction of ~50% of the circulating misinformation.

This seems like a remarkable result, especially in light of the recent revelations in the press about Facebook. The work could have broad impact, and would likely be of interest to an interdisciplinary audience.

There are, however, a number of issues with the manuscript, and I am unable to recommend acceptance of the manuscript as is. Before I can recommend publication in Nature Human Behavior, I hope the authors would consider addressing the following issues:

- To prove their main point, the authors extrapolate from a statistical model without any form of empirical validation. The fact that the model is well calibrated to the original data (without interventions) does not tell us anything about the more important question of whether the same model (with the various interventions) would be a "good" model to describe the same process under those interventions. In this sense, the type of evidence the authors can achieve is purely computational but not causal. I am wondering whether there are ways to provide more causal evidence in support of the main findings (i.e. that combining interventions leads to large reductions in spread), but lacking data about the interventions deployed by the platform (in this case Twitter), it is hard to see how one could go beyond mere speculation guided by a model (albeit a well-calibrated one).

AR: *We agree that this is a key limitation of our approach. However, similar arguments could be put forth against climate models, where we lack empirical evidence that reduction in fossil fuels would have the intended effect. As with climate models, we evaluate our model appropriateness by examining whether our model and fitted parameters can generate data consistent with observations (Fig in SI). Here, we hope that the theory can be evaluated broadly as a tool for deciding how to ethically intervene (and then empirically evaluate the consequences). Indeed we believe that a paper with empirical verification at scale would be more compelling, yet ethically fraught in the absence of broader scientific scrutiny of the theory. We have added some discussion of these limitations.*

R2: - Related to the first point, the authors implicitly assume that the data they observe corresponds to the scenario in which the platform is not making any intervention, however, it is quite likely that Twitter may be already applying some of the interventions studied. In this case, the observed reductions may be biased toward overestimating the amount of reduction one could achieve.

AR: *We have acknowledged this limitation in the discussion and cited some literature on the known interventions applied by Twitter. However, we believe this limitation highlights a broader need for transparency on the part of platforms. We've added discussion to this effect as follows:*

"Moreover, limited transparency regarding interventions applied by Twitter during the election raises the possibility that our model over-estimates the efficacy of interventions that were cryptically in place \cite{Sanderson2021TwitterPlatform}."

R2: - The framing of the results is a bit over-optimistic. True, the findings show that there are alternative methods to extreme censorship (something no platform would consider anyway, at least in the US), however a more pessimistic interpretation of the findings could be that, in order to be effective, interventions will have to necessarily include many different moving parts, making them more complex compared to an approach based on a single intervention.

AR: *We agree and have adjusted our wording throughout. Examples include in the discussion we replaced “fortunately” with “however” and added a paragraph noting limitations of the approach. We agree with the reviewer that a key finding of our paper is that no single solution is likely to be a panacea, something we see as a major contribution of the paper. We hope that our modeling approach (and future work built on it) will highlight the importance of platforms going beyond single solutions and hoping their effects will be sufficient.*

R2: Some other minor issues:

- A couple of passages in the methods section are a bit unclear. In particular I did not understand what is the **baseline rate of detection for related keywords (line 490)** and what is the reason for using linear interpolation for normalization (line 547).

AR: *The linear interpolation for normalization was largely used to present figures resembling a time-series and did not impact the results. We’ve removed the normalized time-series figures and replaced them with simple violin plots.*

R2: - The plots in the supplementary information are too small to be legible. Reducing the number of plots per page could help.

AR: *We’ve removed these yet left them generated by the code if someone seeks to inspect the posterior predictive fits. With our larger dataset, this became a necessity.*

R2: - There are few typos in the manuscript (I spotted some on line 12 (Capitol instead of capital) and 13 (online ... online)), and some sentences were a bit hard to parse (e.g. line 402). I would recommend some more editing especially in the Methods section.

AR: *Noted and revised throughout.*

R2: - The plots are not very well readable in B&W. This is especially the case of Fig 1D, 2, 3, and 4.

AR: *Our revised figures should be more readable in B&W.*

Reviewer #3:

R3: Remarks to the Author:

After multiple readings of this paper I suggest a rejection for Nature Human Behaviour, for the reasons I will list in the following lines. Nevertheless I would like to encourage the authors to add some other huge analysis to their work, I will provide some hints.

AR: *We appreciate the reviewers' suggestions to deep-dive into competing models of information transmission and their implications for the efficacy of interventions. We believe it would make for an interesting paper, particularly given long-standing and largely unresolved debates about the relative utility of competing models of contagious processes across various sociological contexts. For reasons outlined below, however, we believe the "huge" analysis suggested by the reviewer is well beyond—and outside of—the scope of this paper or the limitations of the article formatting requirements for NHB.*

R3: The authors present a new model to measure misinformation spreading and then they consider different policies to reduce misinformation, implementing them in their own model and comparing the results.

R3: First of all, the model lay on highly questionable assumptions. The authors assume that the misinformation spreading follows a "simple contagion" dynamics: this should be better explained and discussed since several works in literature show the opposite thesis. Some references:

- Centola D, Macy M (2007) Complex contagions and the weakness of long ties. *Am J Sociol* 113(3):702–734
- Lerman K (2016) Information is not a virus, and other consequences of human cognitive limits. *Future Internet* 8(2):21
- Romero DM, Meeder B, Kleinberg J (2011) Differences in the mechanics of information diffusion across topics: idioms, political hashtags, and complex contagion on twitter. In: *Proceedings of the 20th International Conference on World Wide Web*. ACM. pp 695–704
- Min B, San Miguel M (2018) Competing contagion processes: Complex contagion triggered by simple contagion. *Sci Rep* 8(1):10422

AR:

We appreciate the reviewer's feedback and the nuanced, important, and long-standing distinction between various contagious processes. We note, however, that we may expect quite different dynamics for the adoption of behaviors (e.g., Centola 2007, Romero 2011, Min 2018) when compared with shorter time-scale decisions to reshare content (Juul 2021). Adoption of behaviors and the like occur over a comparatively longer period of time likely involving much more complex social and cognitive processes.

Of the studies listed above, only the Lerman work examines dynamics on short time scales with the decision to share or reshare content on Twitter (vs. adopt a novel behavior such as joining a health platform or share content on some other platform). Yet this work examines the spread of information prior to Twitter's implementation of algorithmic ranking of content. There is little reason to believe contagious dynamics after the implementation of algorithmic

sorting would be similar to those that occur with chronological timelines. At the very least, a lack of consistency between the two should not be seen as evidence of some flaw with either paper. Indeed it may be an intriguing thread to pull at with the right dataset(s).

Moreover, aspects of the Lerman Twitter results are consistent with simple contagions. Figure 3B's probability of sharing is nearly linear for up to 50 friends Tweeting. This seems fairly consistent with a simple contagion, perhaps with modification to assume some immunity in the population or heterogeneity in exposure (i.e, time on platform). Our decay terms may capture some of this as well. The linearity is consistent with other research showing nearly linear retweet rates with Follower size, a feature of simple contagions/mass action (Martin 2016), research showing that viral spreading online approximates viral contagions (Waang 2011), and quite recent work successfully replicating key findings on the spread of false/true news using disease-like models (2021). Finally, our model incorporates temporal decay (e.g., from new content crowding out old, saturation of susceptible users) which allows for saturation in a way not possible in the ICM model examined by Lerman. In other words, our model relaxes the assumption that contagions are independent, which would dramatically change the results of figure 4 in Lerman.

Overall, our decision to rely on an epistemic (not ontological) assumption of a simple contagion is not particularly novel (Martin 2016, Juul 2021, Wang 2011), nor is it unprincipled for sharing of information on Twitter over short-timescales. We agree with the reviewer that adoption of behaviors (for instance *believing* misinformation) is likely to be a complex contagion perhaps arrived at through exposure to multiple other contagions (Min 2018). However, measuring that at this scale is well beyond the limitations of present knowledge and data access.

Nevertheless, we certainly went to great quantitative lengths to ensure our model is capturing features of the dynamics and is sufficient for our inferential goals. This took two forms, posterior predictive checks, and evaluation of Hamiltonian Monte Carlo diagnostics. The posterior predictive checks---highlighted in the SI, Fig S2, generated in published code---show that our model is not only able to reproduce macroscopic properties of the dynamics (e.g. total exposed) but also captures minute temporal dynamics throughout the course of each information cascade.

If our models' assumptions were dramatically different from the data generating process, we would not expect such a robust fit. Indeed our model inclusion criteria failed to include a minority of detected events as they were not well-approximated by our models' assumptions. Examples of these include narratives with considerable off-platform spread, news coverage, and (perhaps) coordination. One advantage of relying on the Stan implementation Hamiltonian Monte Carlo in this context is that it has a number of built-in checks to ensure that the model is well-specified given the data. We leveraged these heavily in ensuring that we only included events for which our model sufficiently described the dynamics.

Martin, T., Hofman, J. M., Sharma, A., Anderson, A., & Watts, D. J. (2016). Exploring limits to prediction in complex social systems. *25th International World Wide Web Conference, WWW 2016*, 683–694. <https://doi.org/10.1145/2872427.2883001>

Juul, J. L., & Ugander, J. (2021). Comparing information diffusion mechanisms by matching on cascade size. *Proceedings of the National Academy of Sciences of the United States of America*, 118(46). <https://doi.org/10.1073/pnas.2100786118>

Wang, L., & Wood, B. C. (2011). An epidemiological approach to model the viral propagation of memes. *Applied Mathematical Modelling*, 35(11), 5442–5447. <https://doi.org/10.1016/j.apm.2011.04.035>

R3: Second, the users' audience is approximated by their follower count, and this is basically what defines the virality (with the addition of a decay). In my opinion here the authors should better elaborate what is their definition of virality and what they are measuring: virality is something related to influence and influence is a very complex concept and does not necessarily correlates with the number of followers. In other words, the authors propose a model in which a high number of followers lead to high virality, but there are several famous paper that show that this is not what actually happen in social media, or that other factors as limited user attention are very important. Some references:

- Cha, M., Haddadi, H., Benevenuto, F., & Gummadi, K. (2010, May). Measuring user influence in twitter: The million follower fallacy. In *Proceedings of the international AAAI conference on web and social media* (Vol. 4, No. 1).
- Qiu, X., Oliveira, D. F., Shirazi, A. S., Flammini, A., & Menczer, F. (2017). Limited individual attention and online virality of low-quality information. *Nature Human Behaviour*, 1(7), 1-7.

AR: *The editor has asked us not to cite the Qiu paper, which was retracted over code errors. In general, we note that one of the more robust findings in recent years has been the extent to which follower size is a strong predictor of influence (Martin 2016, below). Virality as defined in our model is a latent parameter capturing the momentum of spread at a given point in time. In our paper it takes on a formal mathematical definition which may differ from casual use across the literature (which itself varies widely). We are open to other suggestions for referring to that parameter. The Cha paper represents some fantastic early work on Twitter, although it was conducted well before the implementation of algorithmic filtering which makes its relevance to dynamics over the past few years limited. More recent work has highlighted the appropriateness of simple contagions for evaluating the spread of false information online (Juul 2021).*

Moreover, limited attention may be offset, to some degree, by algorithmic ranking.

Martin, T., Hofman, J. M., Sharma, A., Anderson, A., & Watts, D. J. (2016). Exploring limits to prediction in complex social systems. *25th International World Wide Web Conference, WWW 2016*, 683–694. <https://doi.org/10.1145/2872427.2883001>

Juul, J. L., & Ugander, J. (2021). Comparing information diffusion mechanisms by matching on cascade size. *Proceedings of the National Academy of Sciences of the United States of America*, 118(46). <https://doi.org/10.1073/pnas.2100786118>

Finally, the model does not take into account the network itself, its topology, that actually plays a very crucial role in spreading phenomena.

AR: *As a consequence of algorithmic filtering, the actual network (who sees who) is not publicly available and can only be inferred indirectly. Doing so requires making epistemic commitments regarding what constitutes an edge (i.e., retweeting, retweeting x times, all interactions) and whether/how edges should be weighted. Incorporating the network could easily bias the results in difficult to discern ways. We agree that contagion simulations on Twitter networks would be (and have been) useful, however, we note that our data only represents one possible path of contagion through the network and its utility for network simulations would be limited, further divorcing the model from the data. We additionally note that networks are rarely available for disease transmission, yet models of infectious disease have nonetheless provided a wealth of insight.*

R3: Second, the authors used a dataset to tune the parameters, and then, using the obtained values, they simulated several interventions. Nevertheless, a bit of analytical approach could have been useful to better understand the role of the parameters: I suggest some heatmaps varying two parameters on x, y and measuring in $z(x_i, y_i)$ the averaged engagement over several simulations with those values $x=x_i$ and $y=y_i$.

Then the model could be better understood and maybe compared with the existing ones: about this point, I would also suggest to add a comparison with other existing models and a simple SIR, and proving that this model capture the dynamics better than other models.

AR: *We have added heatmaps of model parameters to the SI, to aid with interpretation. It is unclear to us whether the approach suggested by the reviewer is possible, as our model offers no analytical solution for a given set of parameters. This arises from the fact that simulations/fitting is conditioned on the empirical distribution of follower counts of actual users across time steps. As Bayesian inference generates a joint posterior distribution of parameter estimates, parameter samples from the posterior are proportional to their joint posterior probability given all other model parameters, data, and the likelihood function. Ignoring covariance between parameters and data would produce misleading results.*

Anecdotally, early attempts at modeling the contagion with simpler/alternative models lead us

towards the model we currently employ. Hamiltonian Monte-Carlo is quite sensitive to a poorly parameterized model for a given dataset, particularly for auto-regressive, nonlinear, or hierarchical models. As a consequence, when a model is a poor fit HMC tends to perform poorly—taking a long time to sample and producing incoherent and inconsistent results. Early versions of the model that better approximated a traditional SIR (i.e., no decay term) or non-linearity of the relationship between followers and virality increases resulted in sampling chains taking a long time, exceeding tree-depth, mixing poorly, or failing entirely. Orders of magnitude longer sampling times for an already computationally costly analysis. Incorporating competition and saturation via the decay function alleviated these challenges, suggesting it is hitting on something inherent to the data. Most likely, saturation of the network and competition with newly emerging stories. Nevertheless, model comparison with a model that turns out to be a poor fit to the data is difficult, as the time it takes to fit goes up considerably and the results become much more difficult to interpret.

We also note that a traditional SIR model here provides limited utility and is insufficient given our inferential goals (evaluating interventions). The primary way in which it would deviate from our model is in ignoring or approximating the follower counts of individual users (through r_0/r_t).

This makes it impossible to consider account removal or combined approaches. Other interventions that could be implemented in a SIR context would simply recover well-established properties of SIR dynamics. For instance, VCB as implemented as a decrease in r_t (for example) would reveal well-established relationships between r and total infections. Nudges would recover the consequences of pre-existing immunity in a proportion of the population. In either case, divorcing our simulations/modeling from the core contribution of our paper (conditioning on the full and unique dataset) and would likely push us beyond word limits without allowing direct comparison to our model (as the data cannot be similarly fit) or providing distinct insight into interventions. Further, we note that the monotonically increasing nature of SIR model infections would necessarily lead to worse fit with an SIR model than our model, creating a bit of a strawman.

More generally, we make no claim that our model is a perfect representation of the process, only that it provides a reasonable fit to our data (e.g., posterior predictive checks), samples reliably (HMC diagnostics) and is sufficient to address our inferential goals. The philosophical underpinnings of our modeling approach are highlighted in Betancourt 2020, linked below.

https://betanalpha.github.io/assets/case_studies/principled_bayesian_workflow.html#1_Questioning_Authority

R3: These are the main reasons that made me lean towards rejections. As I mentioned before, in my opinion there are many problems related to the model itself, while the rest of the paper and methodology used to test several interventions are quite ok. It could be interesting to study these interventions also in other proposed models.

Minor issues:

In the model equation, it is not really necessary to define $x(t)$ separately since it can create a bit of confusion.

AR: This parameterization highlights the underlying (auto-) regressive structure of the model. This is somewhat common in descriptions of Bayesian models and matches the parameterization as implemented in the code. Reparameterizations in Stan can affect the performance of the sampler, so our hope here was to be as close to the implementation as possible in the mathematical description. If the other reviewers/editors feel it is appropriate, we are happy to re-parameterize.

Decision Letter, first revision:

14th April 2022

Dear Dr Bak-Coleman,

Thank you once again for your revised manuscript, entitled "Combining interventions to reduce the spread of viral misinformation," and for your patience during the re-review process.

Your manuscript has now been evaluated by Reviewers 1 and 2 from the original round of review. All reviewer feedback is included at the end of this letter. Although the reviewers found your manuscript to have improved during revision, Reviewer 2 raises some remaining concerns regarding your description of interventions as well as the section on Model Validation. We remain very interested in the possibility of publishing your study in *Nature Human Behaviour*, but would like to consider your response to these concerns in the form of a revised manuscript before we make a decision on publication.

In sum, we invite you to revise your manuscript taking into account all reviewer comments. We are committed to providing a fair and constructive peer-review process.

We hope to receive your revised manuscript within 4-8 weeks. I would be grateful if you could contact us as soon as possible if you foresee difficulties with meeting this target resubmission date.

- Include a "Response to the editors and reviewers" document detailing, point-by-point, how you addressed each editor and referee comment. If no action was taken to address a point, you must provide a compelling argument. This response will be used by the editors and reviewers to evaluate your revision.

- Highlight all changes made to your manuscript or provide us with a version that tracks changes.

[REDACTED]

This URL links to your confidential home page and associated information about manuscripts you may have submitted, or that you are reviewing for us. If you wish to forward this email to co-authors, please delete the link to your homepage.

We look forward to seeing the revised manuscript and thank you for the opportunity to review your work. Please do not hesitate to contact me if you have any questions or would like to discuss these revisions further.

Sincerely,

Arunas Radzvilavicius, PhD
Editor
Nature Human Behaviour

REVIEWER COMMENTS:

Reviewer #1:
Remarks to the Author:

This revision has improved the paper substantially. I maintain my original enthusiasm due to the wide scope and high relevance of this work. Now, the new Data and Model Overview section makes it much easier to read the Results section and to focus on a detailed understanding of the model. It seems like a real accomplishment that this relatively simple model can fit so many of the events, plus can be used to simulate interventions.

In fact, the "Response to Reviewers" highlighted a number of strengths of the model (and data) that would be useful to point out in the text. These selling points (for me) include (a) that the model has just five parameters per event (plus uses the initial number of posts and the empirical distribution of followers at each timestep), (b) that the model gives a good fit across orders of magnitude, (c) the comprehensiveness of the data (don't assume the readers will look at the reference), and (d) the argument (from p. 3 of the Response) of how, like with disease spreading models, we know that misinformation is transmitted over a network, yet you've shown we can do a remarkably good job modeling that spread without explicitly using a network -- only the follower counts. (Similarly, a handful of sentences in the center of p. 4 of the Response could [almost!] replace 3-4 paragraphs in the Model Derivation section. I'm exaggerating; but they do make their point effectively.)

(All suggestions in this paragraph are entirely optional.) Since the time this manuscript was submitted, we've seen some astonishing examples of state-sponsored propaganda, plus quick reactions from social media platforms, around Russia's invasion of Ukraine. As it feels like the world has changed dramatically, you might consider updating the introduction. In addition, I wanted to pass along some relevant data points I saw delivered by a Twitter representative. In (webinar) <https://youtu.be/LRUZwX7A1I0?t=470>, they say they're reducing the spread of certain tweets by 80% by doing something like a virality circuit breaker, and then at <https://youtu.be/LRUZwX7A1I0?t=1988>, they say that they've seen pop-up nudges reduce sharing by an average of 40%. Finally, their emphasis on the publishers made me wonder if your account banning and 3 strikes policies could be easily adapted to publishers, as opposed to arbitrary amplifiers of misinformation.

I find the new section on Model Validation fairly unconvincing, due to its focus on the study's limitations. It's worth adding more about the internal validations you have already done (e.g., bottom of p. 10 in Response). I appreciate seeing Fig S3, and I miss the old Fig S2; if there are now too many events to plot legibly, perhaps a random selection could be shown instead. For this section, I was anticipating claims (perhaps along the lines of those in the Principled Bayesian Workflow link) about how well the model reproduces known quantities (perhaps besides total engagement), how consistent or reasonable the inferred parameters are, and perhaps about how the modeled interventions are a good match for what would happen in real life.

On that last item, one point that's not clear to me regards how nudges are simulated. If a pop-up reduces each user's probability of sharing by $x\%$, I can see that in expectation, the number of followers would be reduced by $x\%$. But that doesn't mean the two changes are identical. In particular, I'm concerned about how the follower count contributes to the virality update. Could you clarify how (or if) these two ways to describe the intervention are really the same? Then, separately, could you clarify the intuition for the VCB? Is it that (e.g.) 10% of tweets about an incident are prevented from being amplified, and/or that all tweets about an incident are 10% less likely than usual to be amplified?

Finally, a list of other questions, concerns and typos that shouldn't warrant too much discussion.

-Fig 1: C and D's x-axes are still (again?) different, and all the y-axes should agree on terminology (tweet vs. post vs. engagement). These are all the same event, right?

-(typo) an early reference to Fig 2 omits the word "Supplement."

-Fig S2: shouldn't the lines be horizontal?

-Fig 3D seems to reverse the brown colors from 3B.

-The model equations would be easier to understand (and also match the code) if x_t were always updated before v_t .

-in Model Derivation section: 3rd to last paragraph seems redundant with the next two, and final sentence repeats criteria (a).

-lognormal regression for post-event engagement: please write the equation somewhere so we know what the parameters mean.

-(clarify) The 1504 "currently removed accounts" must mean "by Twitter." Is there a reference for them?

-(clarify) Segmentation of incidents into events: is it that first they're divided by boundaries where the posts dip to $<5\%$ of the max, and then further, any two peaks of $>30\%$ of the max are separated into different events?

-The discussion of during- and post-event engagement naturally brings up questions about pre-event engagement, especially if it's bigger than the other two categories (39%??). The Response addressed this a little.

-Removing verified accounts makes me wonder why you chose those instead of (say) looking for unverified bot accounts. A sentence about the number of followers would probably solve that.

-Regarding the "interventions that were cryptically in place" -- this sentence confused me. "Due to" interventions that were possibly already in place, couldn't your estimates be either too high or too low? (Though either way, your model still estimates the effects of having further added these changes to whatever was already in place.)

Reviewer #2:
Remarks to the Author:

The authors have addressed my comments and I am happy to recommend the article for publication on NHB.

Author Rebuttal, first revision:

Comments to Editor and Reviewer:

Our dataset has gone through some additional curation since we last submitted, these changes were minimal but our overall number of included tweets has increased from ~9.6 to 10.5M. We've updated the manuscript throughout and re-ran the entire analysis. The results have not qualitatively changed and there are indicators that fit has improved.

Reviewer #1:

Remarks to the Author:

This revision has improved the paper substantially. I maintain my original enthusiasm due to the wide scope and high relevance of this work. Now, the new Data and Model Overview section makes it much easier to read the Results section and to focus on a detailed understanding of the model. It seems like a real accomplishment that this relatively simple model can fit so many of the events, plus can be used to simulate interventions.

AR: *We are thankful for the reviewer's engagement with our manuscript and have found reviewer's comments to be particularly helpful in ensuring that our paper is both accessible and convincing to the broad readership of NHB.*

In fact, the "Response to Reviewers" highlighted a number of strengths of the model (and data) that would be useful to point out in the text. These selling points (for me) include (a) that the model has just five parameters per event (plus uses the initial number of posts and the empirical distribution of followers at each timestep), (b) that the model gives a good fit across orders of magnitude, (c) the comprehensiveness of the data (don't assume the readers will look at the reference), and (d) the argument (from p. 3 of the Response) of how, like with disease spreading models, we know that misinformation is transmitted over a network, yet you've shown we can do a remarkably good job

modeling that spread without explicitly using a network -- only the follower counts. (Similarly, a handful of sentences in the center of p. 4 of the Response could [almost!] replace 3-4 paragraphs in the Model Derivation section. I'm exaggerating; but they do make their point effectively.)

AR: *We agree with the reviewer that these would be worth highlighting throughout. We've removed some of the (admittedly length) discussion on challenges to external validation and focused our model validation section more on the strengths of our approach. (Seen in our response to a later comment by R1.*

- a) *We have highlighted that this relies on just 5 parameters, and added posterior predictive time-series of the largest events, to showcase that our model does well even when the number of datapoints far exceeds the number of parameters.*
- b) *We have highlighted the consistency across orders of magnitude more in our model validation section.*
- c) *We have highlighted the comprehensiveness of our dataset at the outset of the results as follows:*
 - i) *Search terms and incidents were identified through real time monitoring and updating by dozens of analysts and several community partners as part of the Election Integrity Partnership \cite{ElectionIntegrityPartnership2021TheElection}. As such, we believe our dataset provides a thorough, if not comprehensive, overview of misinformation during the 2020 US presidential election.*
- d) *We note the argument from the response was in the previous draft section as part of our model derivation as follows, and have brought in some justification from the previous AR as highlighted in blue:*
 - i) *We note that our model does not explicitly incorporate a network, as is common in many simulations of information and behavior spread online \cite{Centola2007c}. Our primary reason for doing this is that algorithmic filtering of content renders the true network topology unknown. Reconstructing a network would require additional epistemic assumptions, which could bias the results in opaque ways \cite{Butts2009RevisitingAnalysis}. Moreover, research on disease has highlighted the utility of modeling interventions in the absence of network structure, notably when the degree distribution is known or approximated \cite{Lloyd-Smith2005SuperspreadingEmergence}. We note the success of simple models in understanding spread of infectious disease is not due to simplistic contagion dynamics. For disease, daily interactions, immune-system dynamics, population structure, behavior, and air-flow patterns create remarkably complex and dynamic network topologies of disease spread.*
- e) *Revised validation here, for convenience:*
 - i) *Here, we take a similar approach to climate models to validate our model internally (i.e., within our dataset). Climate models can be validated by allowing them to condition on data and then run freely for some time period. If the model successfully retrodicts conditions at a future point in time, it provides evidence that the model captures the*

dynamics of interest. We follow much the same approach here, simulating total engagement from the initial tweet throughout an event. At the coarsest level, the total number of observed posts (10.4M) falls within the 89% credible interval of our baseline simulations (10.8M, 89% C.I.: [9.8, 11.7]). On the scale of individual events, simulations recover the number of observed posts over several orders of magnitude, despite the model only being seeded with posts in the first time-step and the time-varying empirical follower distribution (Supplement Fig. \ref{SI-fig:InternalValidation}). This holds true across several orders of magnitude in post volume and for events that vary widely in duration from one hour to several days. Visual inspection of posterior-predictive time series similarly indicates that our model recovers fine-grained temporal dynamics, even for our largest events where the number of datapoints far exceeds model parameters \ref{SI-fig:PosteriorTimeSeries}). Considering the relatively small number of parameters (5 in this model), this provides evidence that our model is adequately capturing key features of the underlying dynamics.

(All suggestions in this paragraph are entirely optional.) Since the time this manuscript was submitted, we've seen some astonishing examples of state-sponsored propaganda, plus quick reactions from social media platforms, around Russia's invasion of Ukraine. As it feels like the world has changed dramatically, you might consider updating the introduction. In addition, I wanted to pass along some relevant data points I saw delivered by a Twitter representative. In (webinar) <https://youtu.be/LRUZwX7A1I0?t=470>, they say they're reducing the spread of certain tweets by 80% by doing something like a virality circuit breaker, and then at <https://youtu.be/LRUZwX7A1I0?t=1988>, they say that they've seen pop-up nudges reduce sharing by an average of 40%. Finally, their emphasis on the publishers made me wonder if your account banning and 3 strikes policies could be easily adapted to publishers, as opposed to arbitrary amplifiers of misinformation.

AR: *Platform responses in the time since writing this have evolved rapidly, and something we have thought about often is whether/when/how to update the manuscript accordingly. Similar things surround the ongoing debate about the efficacy of nudges, particularly across the political spectrum. I am somewhat hesitant to cite the internal twitter research on the efficacy of their interventions, given the lack of independent ability to replicate their findings. Moreover, we indeed can't cite these sources given NHB publishing guidelines. However, I would be *very* interested—for instance—in knowing which subset of information that 40% nudge impacts and among whom. At the end of the day, one of our rationales for sharing the code and data (beyond obvious transparency) was to enable other researchers to try out different parameters, interventions, etc...*

On publishers, our approach doesn't distinguish between them and other accounts explicitly although they do tend to have larger followings than the average user. Although the model would require some retooling to incorporate sources of claims, I do imagine that reducing publishers (here sources) would have an outsized effect compared to amplifiers. This is something I've thought about but just simply haven't gotten around to as it would require a bit of manually coding accounts in our dataset. Perhaps it would make a good follow-up paper...

I find the new section on Model Validation fairly unconvincing, due to its focus on the study's limitations. It's worth adding more about the internal validations you have already done (e.g., bottom of p. 10 in Response). I appreciate seeing Fig S3, and I miss the old Fig S2; if there are now too many events to plot legibly, perhaps a random selection could be shown instead. For this section, I was anticipating claims (perhaps along the lines of those in the Principled Bayesian Workflow link) about how well the model reproduces known quantities (perhaps besides total engagement), how consistent or reasonable the inferred parameters are, and perhaps about how the modeled interventions are a good match for what would happen in real life.

AR: *The old Fig S2 has returned (as Fig S3). We love these plots and agree that they really highlight how the dynamics are being captured by the model. As you intuited, our larger dataset made it impossible to add 1000+ plots to the SI and have the pdf both open and be legible. Instead, we've included the hardest case for our model—the largest and longest events. We have edited the end of our model validation section to more firmly hit the points highlighted by the reviewer and the reason we have faith that our model is informative.*

- *Here, we take a similar approach to climate models to validate our model internally (i.e., within our dataset). Climate models can be validated by allowing them to condition on data and then run freely for some time period. If the model successfully retrodicts conditions at a future point in time, it provides evidence that the model captures the dynamics of interest. We follow much the same approach here, simulating total engagement from the initial tweet throughout an event. At the coarsest level, our model successfully reproduces the observed patterns of total engagement across several orders of magnitude (Supplement Fig. \ref{SI-fig:InternalValidation}). Visual inspection of posterior-predictive time series similarly indicates that our model recovers fine-grained temporal dynamics, even for our largest events where the number of datapoints far exceeds model parameters \ref{SI-fig:PosteriorTimeSeries}). Considering the relatively small number of parameters (5 in this model), this provides evidence that our model is adequately capturing key features of the underlying dynamics.*

On that last item, one point that's not clear to me regards how nudges are simulated. If a pop-up reduces each user's probability of sharing by $x\%$, I can see that in expectation, the number of followers would be reduced by $x\%$. But that doesn't mean the two changes are identical. In particular, I'm concerned about how the follower count contributes to the virality update. Could you clarify how (or if) these two ways to describe the intervention are really the same? Then, separately, could you clarify the intuition for the VCB? Is it that (e.g.) 10% of tweets about an incident are prevented from being amplified, and/or that all tweets about an incident are 10% less likely than usual to be amplified?

AR: *We describe how we implemented nudges mathematically in the Computational model section (below for convenience). I think you're correct that they are the same in expectation although subtly different. Our framing takes the expectation (making them the same) but an alternative approach would be to have the number of followers be distributed binomially ($n, 1-x$). This would add a bit of variance to the total number of followers reduced and *could* have effects, particularly for small cascades and early*

on in the time-series. I don't believe this is likely to meaningfully impact our results, given the amount of engagement driven by large accounts and large events.

In our model, nudges linearly decrease the "boost" to virality which is a linear function of the sharing account's followers. This differs from the virality circuit breaker, which we implement as a proportional reduction in virality at each time step after a fixed period of time. With virality being a latent parameter, it effectively "stores" information about the history beyond a single time step, and VCBs reduce the latent parameter directly in its entirety rather than simply reducing the rate at which it grows. VCBs in a sense "flush out" past contributions to virality, acting more like nonlinear decay. A crude analogy might be vaccines (nudges) vs masks (VCBs). Vaccines reduce susceptibles, masks reduce transmission.

- *Nudge: We implemented nudges via multiplying follower counts by a constant, reducing the pool of susceptible accounts (i.e., for account j , $\hat{F}_j = F_j (1 - \eta)$).*
- *VCB: For example, a 10% reduction in virality was implemented as $\hat{v}_t = v_t (1 - 0.1)$.*

Finally, a list of other questions, concerns and typos that shouldn't warrant too much discussion.

-Fig 1: C and D's x-axes are still (again?) different, and all the y-axes should agree on terminology (tweet vs. post vs. engagement). These are all the same event, right?

AR: Fixed, these are the same event.

-(typo) an early reference to Fig 2 omits the word "Supplement."

AR: Fixed

-Fig S2: shouldn't the lines be horizontal?

AR: Good catch, we've swapped the axes (horizontal looked strange, for some reason)

-Fig 3D seems to reverse the brown colors from 3B.

AR: Fixed

-The model equations would be easier to understand (and also match the code) if x_t were always updated before v_t .

AR: Change adopted

-in Model Derivation section: 3rd to last paragraph seems redundant with the next two, and final sentence repeats criteria (a).

AR: Removed and integrated

-lognormal regression for post-event engagement: please write the equation somewhere so we know what the parameters mean.

AR: *Added*

-(clarify) The 1504 "currently removed accounts" must mean "by Twitter." Is there a reference for them?

AR: *Clarified: "These accounts were identified by examining accounts with posts in our dataset that could not be retrieved with an API call in late January."*

-(clarify) Segmentation of incidents into events: is it that first they're divided by boundaries where the posts dip to <5% of the max, and then further, any two peaks of >30% of the max are separated into different events?

AR: *Clarified discussion of segmentation to highlight that we first identify peaks, then split events using quiescent periods, even if multiple peaks fall into the same event. We repeat this process sequentially until all peaks >30% have become part of an event.*

-The discussion of during- and post-event engagement naturally brings up questions about pre-event engagement, especially if it's bigger than the other two categories (39%??). The Response addressed this a little.

AR: *As we mention in the text "within-event" engagement used to predict post-event engagement is only looking at the largest event per incident. A fair chunk of "pre-event" engagement is going to be smaller events preceeding a larger event, or long amounts of time where low-level noise accumulates before the event goes viral. We've clarified this in the methods, where we've additionally added the request formula.*

-Removing verified accounts makes me wonder why you chose those instead of (say) looking for unverified bot accounts. A sentence about the number of followers would probably solve that.

AR: *Verification is a clear observable that is reported by the Twitter API. It would be much harder to (reliably) label bots in our dataset. We considered a bot removal section and realized it may tell us more about our bot detection algorithm of choice than the benefit of removing of bots. Another reason for considering verified users is that during the Election Integrity Partnership, we noticed considerable amplification by large, verified accounts.*

-Regarding the "interventions that were cryptically in place" -- this sentence confused me. "Due to" interventions that were possibly already in place, couldn't your estimates be either too high or too low? (Though either way, your model still estimates the effects of having further added these changes to whatever was already in place.)

AR: *Rephrased*

- *Moreover, limited transparency regarding interventions used by Twitter makes it possible that some of the simulated interventions were in place, and our simulations reveal the benefit of*

increasing those interventions beyond their implemented amount
{cite{Sanderson2021TwitterPlatform}}

Reviewer #2:

Remarks to the Author:

The authors have addressed my comments and I am happy to recommend the article for publication on NHB.

This email has been sent through the Springer Nature Tracking System NY-610A-NPG&MTS

Decision Letter, second revision:

Our ref: NATHUMBEHAV-210916557B

19th April 2022

Dear Dr. Bak-Coleman,

Thank you for submitting your revised manuscript "Combining interventions to reduce the spread of viral misinformation" (NATHUMBEHAV-210916557B), and for addressing the remaining concerns raised by Reviewer 1. We will now be happy in principle to publish it in Nature Human Behaviour, pending minor revisions to comply with our editorial and formatting guidelines.

We are now performing detailed checks on your paper and will send you a checklist detailing our editorial and formatting requirements within a week. Please do not upload the final materials and make any revisions until you receive this additional information from us.

Sincerely,

Arunas Radzvilavicius, PhD
Editor
Nature Human Behaviour

Final Decision Letter:

Dear Dr Bak-Coleman,

I am pleased to inform you that your Article "Combining interventions to reduce the spread of viral misinformation", has now been accepted for publication in Nature Human Behaviour.

Please note that *Nature Human Behaviour* is a Transformative Journal (TJ). Authors whose manuscript was submitted on or after January 1st, 2021, may publish their research with us through the traditional subscription access route or make their paper immediately open access through payment of an article-processing charge (APC). Authors will not be required to make a final decision about access to their article until it has been accepted. IMPORTANT NOTE: Articles submitted before January 1st, 2021, are not eligible for Open Access publication. Find out more about Transformative Journals

We welcome the submission of potential cover material (including a short caption of around 40 words) related to your manuscript; suggestions should be sent to Nature Human Behaviour as electronic files (the image should be 300 dpi at 210 x 297 mm in either TIFF or JPEG format). Please note that such pictures should be selected more for their aesthetic appeal than for their scientific content, and that colour images work better than black and white or grayscale images. Please do not try to design a cover with the Nature Human Behaviour logo etc., and please do not submit composites of images related to your work. I am sure you will understand that we cannot

make any promise as to whether any of your suggestions might be selected for the cover of the journal.

With best regards,

Arunas Radzvilavicius, PhD
Editor
Nature Human Behaviour